# A new multi-method approach for dating cave calcite: application to the cave of the Trou du Renard (Soyons, France)

Loïc Martin[1,2], Julius Nouet[3], Arnaud Dapoigny[1], Gaëlle Barbotin[2], Fanny Claverie [2], Edwige Pons-Branchu[1], Jocelyn Barbarand[3], Christophe Pécheyran[2], Norbert Mercier[4], Fanny Derym[5], Bernard Gély[6], Hélène Valladas[1],

[1]LSCE/IPSL, UMR 8212, CEA-CNRS-UVSQ, Université Paris-Saclay, Chemin de Saint Aubin - RD 128, F-91191 Gif sur Yvette Cedex, France
[2]Université de Pau et des Pays de l'Adour, E2S UPPA, CNRS, IPREM, Avenue de l'Université, BP 576 64012 PAU cedex, France
[3]Géosciences Paris Saclay, Université Paris-Saclay, CNRS, bâtiment 504, 91405 Orsay, France
[4]Archéosciences Bordeaux, UMR 6034 CNRS - Université Bordeaux Montaigne
Maison de l'archéologie, Esplanade des Antilles, 33607 PESSAC Cedex
[5]Site archéologique de Soyons, 28 rue de l'église, 07130 Soyons
[6]Service Régional de l'Archéologie de la région Rhône-Alpes, 6 quai Saint Vincent 69283 Lyon cedex

*Correspondence to*: Loïc Martin (loic.martin@glasgow.ac.uk)

**Abstract.** A multi-method approach aimed at characterizing carbonate parietal deposits and at proposing a chronology for these carbonate crusts is described. Dating was performed by radiometric methods (C-14 for recent samples, and U-series) on samples that had been characterized beforehand using optical and cathodoluminescence microscopy, and Fourier Transform Infrared microspectroscopy. For U-series, high precision on U-Th ages was achieved using liquid phase multicollector-ICP-MS applied to large samples, while laser-ablation single collector - ICP-SFMS provided information on the reliability of the sampling with a high spatial resolution. This methodology, based on the combination of these two techniques reinforced by the information obtained by the calcite characterization methods, was applied to carbonate deposits from the cave of the Trou du Renard (Soyons, France). The ages obtained with the two U-Th dating techniques are comparable and illustrate that different laminae were deposited at different rates in the samples. In the future, this procedure based on the mineralogical and geochemical characterization of the samples and their dating by radiometric methods will be applied to the layers of parietal carbonates deposited on Palaeolithic decorated walls. When the crystallization is slow, the U/Th dating method by imaging technique is of interest as well as that by multicollector-ICP-MS in liquid phase. The development of robust dating methods on very small quantities of material will make it possible to define the chronological framework of cave rock art.

## 1 Introduction

Establishing the chronology of cave art is a fundamental objective of studies on the cultural evolution of past societies. In Western Europe, and particularly in France, decorated caves are abundant but most of them have yielded figures consisting of engravings or made with metal oxides that cannot be dated directly. Nevertheless, the fact that many decorations are covered

with carbonate deposits that can be dated by radiometric methods opens a new field of investigation for research on cave art since these deposits can, if they are not too old, be dated by the 14C method (Sanchidrian et al. 2017) but also by the U-series method (Pike et al, 2012; Aubert et al., 2014; Hoffmann et al., 2017) as well as by a combination of the two (Valladas et al., 2017). However, with the U-series method, the main difficulty lies in the possible opening of the geochemical system, in the event of detrital contamination by the surrounding sediments during the carbonate formation, or in the event of carbonate alteration accompanied by leaching of uranium (Perrin et al., 2014; Scholz et al., 2014, Pons-Branchu et al., 2020). In order to evaluate these phenomena that can affect carbonate deposits, and hence the U-Th ages, our objective here was to set up an experimental procedure combining different characterization and dating methods. We first tested this procedure on a cavity without any decorated elements where sampling could not deteriorate rock art.

The procedure must necessarily consider two important points. Firstly, in the context of prehistoric cave art, which is subject to preservation imperatives, it is necessary to be able to minimize the impact of the study, and therefore to optimize the sampling, while seeking to obtain a maximum of information. Secondly, to ensure the reliability of the results, it is important to assess the geological and archaeological context, which requires knowledge of the nature of the carbonate mineral and its evolution over time. For this purpose, we undertook petrographic analysis including optical and cathodoluminescence (CL) microscopy, and Fourier Transform Infrared microspectroscopy (μ-FTIR).

To test the relevance of this approach, we sampled two fragments of carbonate covering the wall of the Pillar Room in the "Trou du Renard" cave (Caves of Soyons, France), and a fistula fragment from the same room. Firstly, the petrography of each sample was determined. Secondly, quantitative multi-element mapping using a femtosecond laser ablation system coupled to a single collector sector field ICPMS (fsLA-single collector-ICP-SFMS), was carried out. High spatial and temporal resolutions were achieved on several samples and sub-samples, thus improving the understanding of calcite deposition on the cave walls. Subsequently, the carbonate samples were dated by the U-series disequilibrium method (U-Th) using two techniques: 1) Liquid phase - Multi Collector – Inductively Coupled Plasma Mass Spectrometry (Liq-MC-ICPMS) on dissolved and purified samples (following Pons-Branchu et al. 2020) and 2) a more recent approach based on U-series disequilibrium imaging using fsLA-single collector-ICP-SFMS (following Martin et al., 2022). The 14C dating method using the accelerator mass spectrometry (AMS) technique was also used for the most recent sample. This experimental procedure, combining the characterization of the nature of the carbonate itself and of its constitution, made it possible to obtain information on its temporal evolution, to compare the dating results and to discuss their reliability. This approach thus highlights the contribution of each method, as well as the advantage offered by their combination.

## 2 Materials and methods

### 2.1 From the sampling in situ to sample processing

The Trou du Renard cave, located in the southeast of France in the middle part and on the right bank of the Rhone valley, is composed of two main galleries: the Double-Borne network, where all the studied samples were taken from, and the Ursus

network. It is part of a Jurassic Kimmeridgian karstic network presenting six other caves including the Moula-Guercy cave (Cailhol & Audra, 2013), which is well known for the unearthing of remains of cannibalized Neanderthals (Defleur et al., 1999). Although no human remains were found in the Trou du Renard cave, paleolithic fauna remains are present as well as two Mousterian flint flakes discovered in the Ursus network (Argant, 2010).

Three calcite samples from the Trou du Renard cave were analysed. The first sample (SOY19-01) is a fistula fragment that

fell naturally from the ceiling of the Pillar Room between 2009 and 2010 (Fig 1.a). The second and third samples, designated SOY19-02 and SOY19-03 respectively, are carbonate layers deposited on the wall of the Pillar Room. The sampling site corresponds to a place in the wall that is partially broken, where work has been undertaken to join a lower gallery (Fig 1.b). Sample SOY19-02 fell easily to the ground during sampling. Sample SOY19-03 was manually detached from the wall using a small electric saw, right next to the position of SOY-19-02. Given the proximity of these two samples (SOY19-02 and

SOY19-03), they can be considered to represent the same depositional levels and, in each of them, two carbonate layers separated by a clay film were distinguished. The resulting sub-samples were then designated as SOY19-02 Exo and SOY19-03 Exo for the external parts (surface side), and SOY19-02 Endo and SOY19-03 Endo for the internal part (Fig 1.c). We decided to concentrate our efforts on a single sample and to perform most of our investigations on SOY19-02 and only a few on SOY19-03 in order to compare the respective characteristics of the two samples. Sample SOY19-01 and the two sub-

samples of SOY19-02 were then split into several pieces to allow for the different analyses.

**Figure 1: Samples SOY-19-02 and SOY-19-03**

**(a) view of the area where SOY-19-01 was found (photographs by Damien Butaeye)**

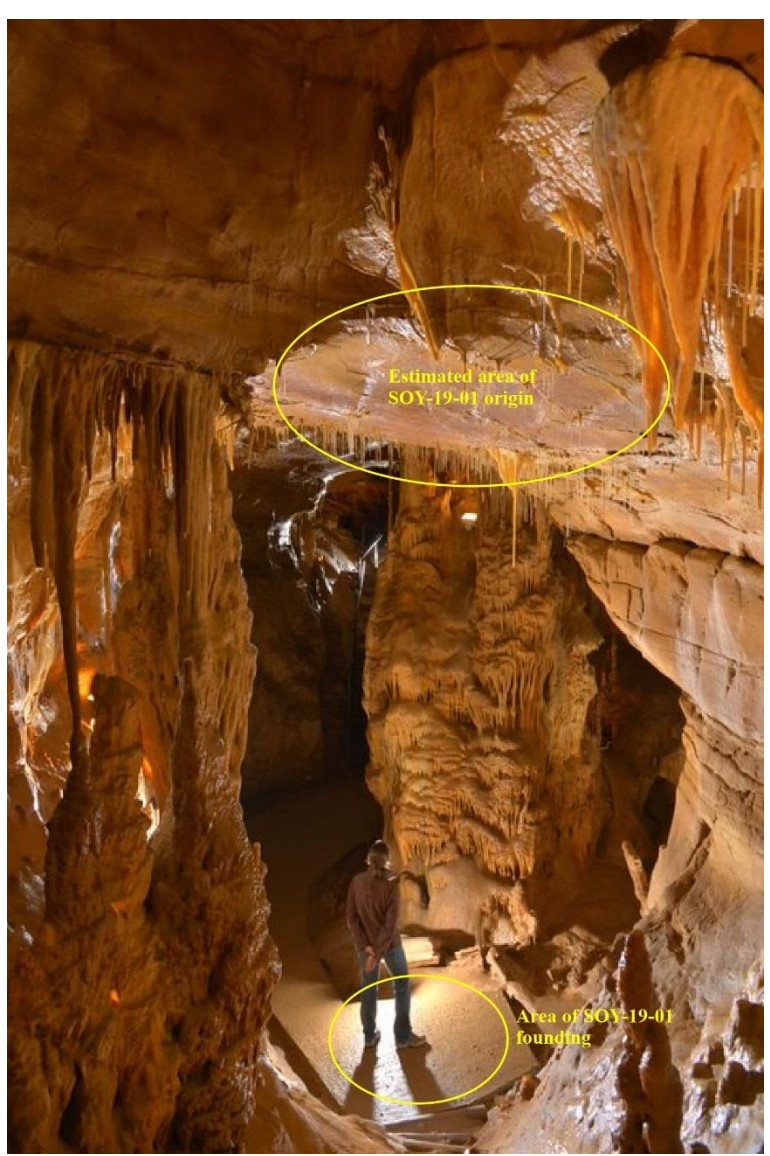

**(b) view of the sampling area of SOY-19-02 and SOY-19-03.**

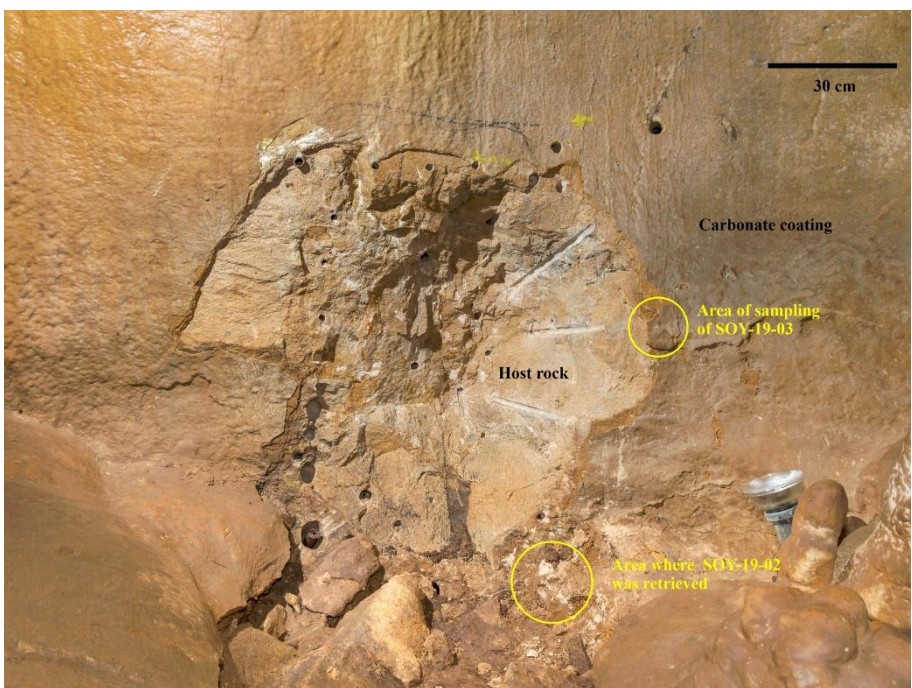

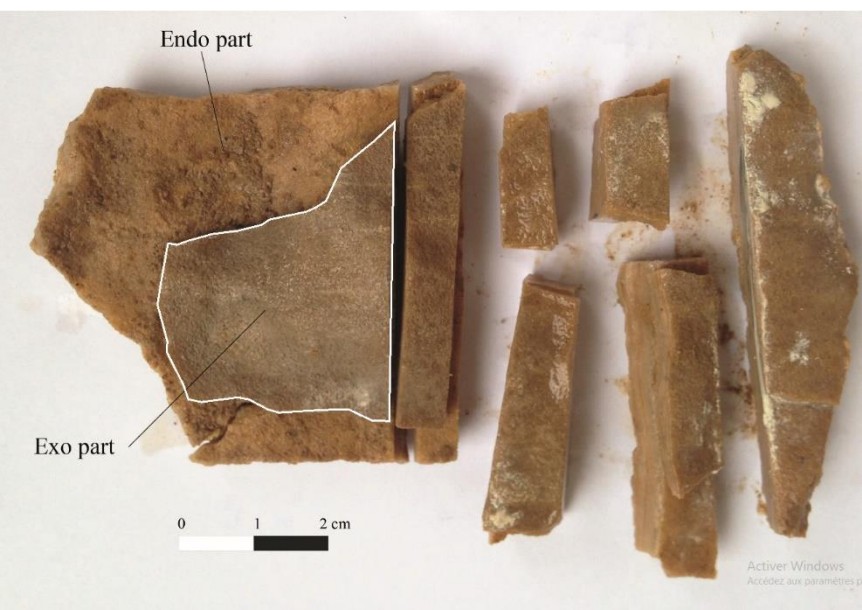

**(c) preparation of sample SOY-19-02 for analyses. The white line indicates the limit between the endo and exo parts.**

## 2.2 Petrographic analysis

Each sample was prepared for the realization of thin slides and epoxy resin blocks. Slides were observed by optical and cathodoluminescence microscopy, while resin blocks were used for FTIR imaging after polishing. Microscopic images of the

samples were acquired using a Leica microscope under plane polarized light (PPL) at a magnification of x25. Cathodoluminescence (CL) imaging was carried out using a Newtec cold cathode system operating at ~10 kV and ~250 μA coupled to a BX41 Olympus microscope at a magnification of x25. The images were modified using retouching software to increase contrast and colour.

Fourier Transform-Infrared images were acquired using a Spotlight 400 FT-IR imaging system coupled with a Frontier IR spectrometer, both from Perkin Elmer, in reflection mode on polished surfaces. Rapid scans of SOY19-02 and SOY19-03 confirmed the high similarity between these samples. High resolution FT-IR scans of the sub-samples Endo and Exo of SOY19-02 were then performed. Four absorption bands are observable in carbonates: a broad high intensity band (1500-1400 $cm^{-1}$), two broad medium intensity bands (1100-1000, 900-800 $cm^{-1}$) and a very weak absorption band near 700 $cm^{-1}$ (Andersen and

Brečević, 1992). The band at 900-800 is called v2 and the band near 700 $cm^{-1}$ is called v4. The v4 band and the band at 1500-1400 $cm^{-1}$ are split into two smaller bands. The position of the v2 band distinguishes the different carbonate minerals: calcite (874-878 $cm^{-1}$), aragonite (853-859 $cm^{-1}$), dolomite (881 $cm^{-1}$) and magnesite (887 $cm^{-1}$) (Huang and Kerr, 1960; Andersen and Brečević, 1992 and references therein). The position of these bands is indicative as the data produced in this study correspond to reflection bands.

## 2.3 Mapping of elements

After petrographic analysis, the resin block samples were recut to fit the femtosecond laser ablation cell, and the sections were polished in order to obtain a flat surface for ablation. Elemental mappings via $^{238}U$, $^{232}Th$, $^{27}Al$, $^{24}Mg$ and $^{43}Ca$ isotopes were obtained using high repetition rate 257 nm UV femtosecond Laser Ablation (Lambda 3, Nexeya, Pessac, France) coupled to a single collector sector field ICP-MS fitted with a jet interface (Element XR, Thermo Fisher Scientific$^{TM}$). The samples were

ablated at a repetition rate of 1000 shots per second using a combination of two simultaneous movements: on the one hand, a vertical back-and-forth laser beam movement of 50 μm at a speed of 1 $mm.s^{-1}$ provided by two galvanometric scanners fitted into the laser machine and on the other hand, the movement of the X-Y stage supporting the sample consisting in successive lines with 50 μm spacing at a speed of 50 $μm.s^{-1}$. The number of counts per mass was read per cycle of 1 s, resulting in a final reconstructed image of 50 μm square pixels. Image sizes ranged from 5.5 mm to 8.4 mm, and each took from 4 h to 6 h of

measurement. $^{238}U$ was mapped in order to investigate the uranium distribution in the samples, while $^{232}Th$ mapping was used to evaluate the potential detrital correction that needed to be applied to U-series disequilibrium dating. The $^{27}Al$ and $^{24}Mg$ elements were considered as proxies of clay or sediment deposit in the samples. $^{43}Ca$ was measured as the proxy of carbonate in the sample.

### 2.4 U-Th dating methodology

#### 2.4.1 Liq-MC-ICPMS

The complete protocol used to prepare the samples, measure isotopic ratios and calculate ages is described in Pons-Branchu et al. (2014). Sub-samples of SOY19-01 and SOY19-02 Exo and Endo were cut in the laboratory using a rotary micro-saw. The Endo and Exo parts were separated manually and their surface cleaned to remove clay deposits. Two samples (int and ext, respectively corresponding to the first third and to the last third of the sample starting from the centre, see Fig.B1) were extracted from SOY 19-01, and two (Endo 1 and Endo 2, respectively corresponding to the first third and to the last third of the sample starting from the base, see Fig.B2) from the SOY 19-02 Endo sample. A piece of SOY19-02 Exo, containing all the micro-layers that are highlighted by the imaging methods, was retained without further subsampling and will be referred to as SOY19-02 Exo "bulk" in the results. The different sub-samples were weighed (between 68 mg and 167 mg) in PFA Teflon™ beakers where a known amount of $^{229}$Th-$^{236}$U spike calibrated against the HU-1 uraninite assumed to be at secular equilibrium, had been previously added. Then the separation of U and Th fractions was performed on U-TEVA resin in nitric media. The isotopic ratios were measured with a Thermo Scientific™ Neptune™ Plus fitted with a jet pump interface and an Aridus II desolvating system.

#### 2.4.2 fsLA-single collector-ICP-SFMS

The resin indurated samples were used for direct U-Th dating by high-sensitivity and high-resolution fsLA-single collector-ICP-SFMS mapping of $^{238}$U, $^{234}$U, $^{230}$Th and $^{232}$Th. The details of the dating protocol are described in Martin et al. (2022). Mappings of SOY19-01, SOY19-02Endo and SOY19-02Exo were performed with sizes of 5.4x1.35 mm$^2$, 4.2x4.8 mm$^2$, and 2.79x3.3 mm$^2$ respectively. This corresponds to a sample mass of approximately 1 to 3 mg per mapping. A 10 µm laser beam diameter delivered at a repetition rate of 1 kHz was continuously and rapidly moved (1 mm.s$^{-1}$) according to a vertical back and forth movement of 40 µm while the sample was moved horizontally at a speed of 50 µm.s$^{-1}$, resulting in 50 µm wide ablation lines. The accumulated counts were read every 1 s. This resulted in mappings with a resolution of 50 µm in both X and Y direction. Each mapping includes the measurement of a blank for 30 minutes before and after sample ablation. A High Resolution ICPMS (Element XR, Thermo Fisher Scientific™) fitted with the Jet Interface was used for detection. The laser was coupled to the ICPMS in a wet plasma configuration by means of a modified three-inlet cyclonic spray chamber that allows the dry aerosol from the ablation cell to be mixed axially upstream of the injector with the wet aerosol from the nebulization. The third inlet, tangential to the position of the nebulizer, was used to introduce a 10 ml nitrogen flow into the argon stream to obtain the best performance from the jet interface option. During ablation, a solution containing 2% HNO3 diluted in ultra-pure water was nebulized in the spray chamber, while during the mass bias calibration procedure, the laser was stopped and the USS was nebulized in the spray chamber The fsLA-ICPMS coupling was tuned daily with a NIST 612 glass sample in order to obtain the best sensitivity while ensuring complete atomisation of the particles. This was achieved by checking that the value of the U/Th ratio measured on the NIST 612 corresponded to the reference value of 1.00±0.05 at 95%

confidence level (95% CL). As laser ablation is less efficient on glass than on calcite, it is assumed that the U/Th elemental fractionation was also negligible during calcite ablation. This stoichiometric detection of U and Th was checked prior to each image acquisition. fsLA allows the use of liquid standard for calibration. A U standard solution (USS) of 0.02 μg·L$^{-1}$ was used for the calibration of the measurement. It was prepared from IRMM 184 SRM (IRMM, Geel, Belgium) in 2% HNO3 (Ultrex, Baker) diluted in ultrapure water (Milli-Q, Millipore) with 0.1% CaCO3 (Suprapur, Merck Darmstadt, Germany). This solution contained a certified $^{235}U/^{238}U$ isotope ratio of $(7.2623 \pm 0.0022) \times 10^{-3}$ at 95% CL and was used to correct mass bias. U/Th fractionation was tested with fsLA and found to be negligible (Martin et al., 2022). Spatial variations in $^{238}U$ content and $^{232}Th/^{238}U$ ratio were used to define several "*Regions Of Interest*" (ROI) on the SOY19-02 mappings, potentially corresponding to different periods of calcification. The isotope ratios used for U-Th dating were calculated for each of the ROIs.

### 2.4.3 Detrital corrections

Two approaches were used for U-Th age corrections. The first one was based on an *a priori* $^{230}Th/^{232}Th$ value for the detrital fraction, here an activity ratio of $1.50 \pm 0.75$ at 95% CL. This value of 150 has been identified as the median value for the $(^{230}Th/^{232}Th)A_0$ of the detrital phase for the dating of speleothems (Hellstrom et al., 2006) and is a commonly used value for speleothem age correction (Martín-García., 2019, Genuite et al., 2022, Pons-Branchu et al., 2022). The second approach was used for the results obtained using fsLA-single collector-ICP-SFMS. Since this method enables several ages to be obtained on the same section, with clear stratigraphic positions, the corrected ages can be modelled using stratigraphic constraints: a large range of $^{230}Th/^{232}Th$ values for the detrital fraction is tested for age corrections, and the model keeps those that give ages in stratigraphic order after correction. Here we used the STRUTAge routine (Roy-Barman and Pons-Branchu, 2016). The "STRUTAge routine" consists of a script, available as supplementary material in the initial article, that can be used with Gnu Octave freeware (http://www.gnu.org/software/octave/). Basically, this routine combines stratigraphical constraints as proposed by Hellstrom 2006 and coevality constraints as in the isochronal approach, but without requiring a single $(^{230}Th/^{232}Th)A_0$ initial. This method tests a large range of correction (Monte Carlo simulation) for detrital thorium and gives the best estimate of the initial $^{230}Th/^{232}Th$ ratio and the corrected age of each sample. With this routine, the variability of the $^{230}Th/^{232}Th)A_0$ can be fixed (details of the parameters fixed for the studied samples are given under the table of results).

### 2.5 $^{14}C$ dating

A sub-sample of SOY-19-01 of about 10 mg was taken using a rotating micro-saw for $^{14}C$ analysis. It was hydrolysed with H$_3$PO$_4$ to obtain CO$_2$ and converted to graphite (Dumoulin et al., 2017) for measurement at the Artemis AMS-French National facility (CEA Saclay, LMC14; Moreau et al., 2020). Carbon isotope ratios were corrected for isotopic fractionation based on δ$^{13}C$ values measured on the AMS, following international recommendations (Mook and van der Plicht, 1999). The $^{14}C$ results were calibrated using Intcal20 (Reimer et al. 2020) with no correction for the proportion of dead carbon (DCP), and with corrections for different DCP (5%, 10% and 20%, which represents the common range of DCP within speleothems, see table 2), following Sanchidrian et al. (2017). They are presented in table 2.

## 3 Results

### 3.1 Petrography and mapping results

The section of the fistula shows a radial structure with two homogeneous crystallization domains: a central dark domain and a peripheral main domain consisting of successive layers of carbonate (Appendix A, Fig A1). The passage between the two domains is marked in CL by a red band while the rest of the sample is non-luminescent. The mineralogy is homogeneous with a position of the v2 band characteristic of poorly crystallized (Full Width at Half Maximum: FWHM >10 cm$^{-1}$) calcite (875-878 cm$^{-1}$).

The carbonate levels sampled for SOY-19-02 and SOY-19-03 consist of thin layers that appear very dark under optical microscopy, alternating in places and mainly in the outer part with layers made of larger crystals (Fig 2.a and Appendix A, Fig A2.a). The growth is quite regular and shows well developed carbonate veils. The CL image shows a high degree of homogeneity for the endo sample, with luminescence in the blue hues characteristic of low-luminescent carbonates. The luminescence is more variable for the exo sub-sample, showing some more luminescent levels, mainly towards the outside,

with a red luminescence hue. These levels are relatively continuous and can be interpreted as the consequences of a change in water chemistry or the incorporation of trace elements into the carbonates, such as Mn, which is the main activator of carbonate luminescence (Machel et al., 1991).

**Figure 2: Images of the SOY-19-02 sample in optical PPL microscopy and CL microscopy. Magnification is x25.**

**(a) optical PPL microscopy**

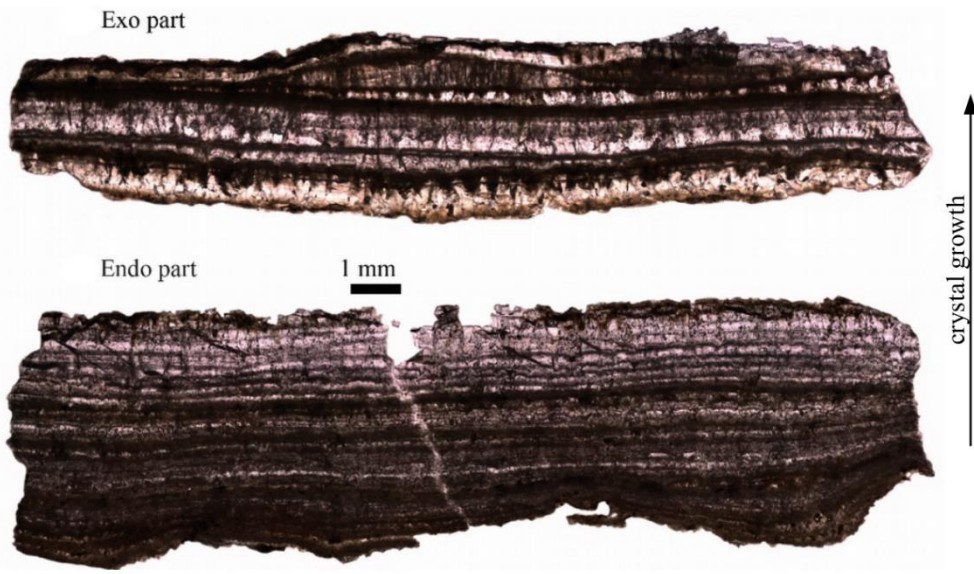

**(b) CL microscopy**

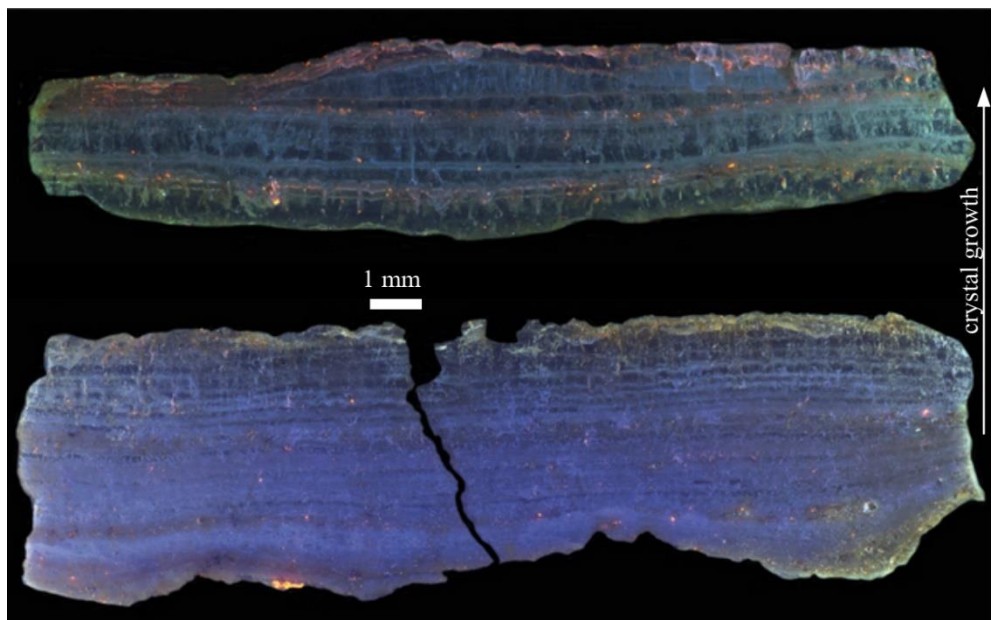

**Figure 3: FT-IR analysis of SOY-19-03. The position of the v2 band is represented on the left while the full width at half-maximum of the v2 band is on the right. The Exo part is at the top of the images.**

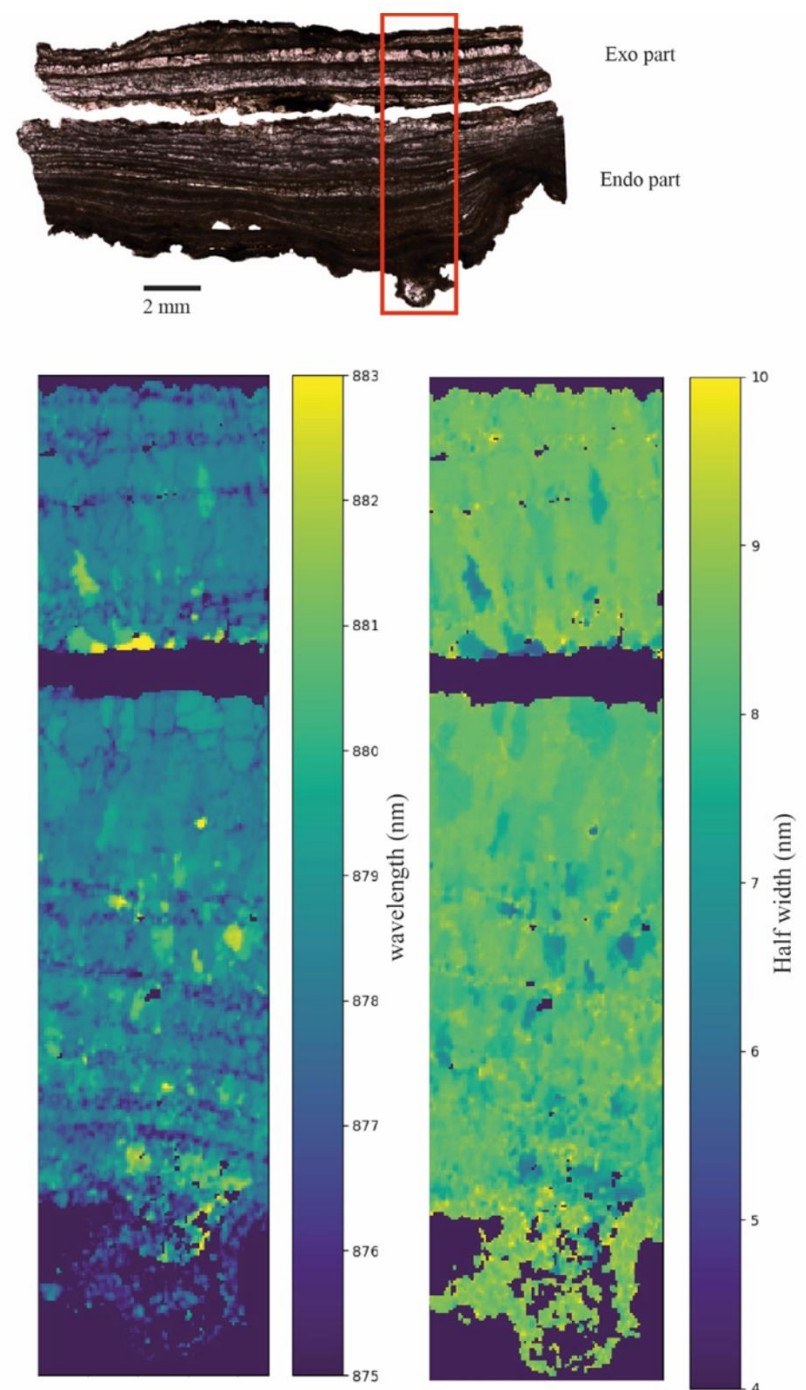

The FT-IR image shows that most of the sample is very homogeneous and consists of calcite crystals ($v2$ band between 876 and 880 cm$^{-1}$) which are well crystallized (FWHM < 10 cm$^{-1}$) (fig. 3). Some minor components show a different FT-IR signal

with a higher wavenumber, especially at the interface between the Endo and Exo sub-samples (Fig 3). This difference in
spectroscopic signatures may be linked to divalent cation substitution. A similar but larger displacement of the v2 band is
indeed interpreted as an increase in substituted Mg along the solid solution between calcite, dolomite and magnesite (Huang
and Kerr, 1960).

A large Al and Mg rich zone is noticeable at the root of SOY19-02 Endo (at the top of the images) on the fsLA-single collector
ICP-SFMS mappings, which corresponds to a piece of the limestone host rock mixed with calcite deposit. Apart from this
basal part, the fsLA-single collector ICP-SFMS mappings of SOY19-02 Endo indicate a significantly more homogeneous
distribution of the chemical elements investigated ($^{24}$Mg, $^{27}$Al, $^{238}$U, $^{232}$Th and $^{43}$Ca) than SOY19-02 Exo, which shows
standard deviations between pixel values 1.3 to 3 times higher than SOY19-02 Endo.(Fig 4). The mappings highlight the more
pronounced presence of detrital layers between the calcite layers in SOY19-02 Exo, identified in particular by higher Al and
Mg contents. The $^{238}$U and $^{232}$Th values show a good correlation with the presence of detrital layers. This was confirmed by
the additional measurement of uranium and thorium isotopes required for U-Th dating and was considered by correcting the
ages from the detrital $^{230}$Th fraction (Table 1, Fig 6).

**Figure 4: FsLA-quad-ICPMS qualitative mappings of $^{24}$Mg, $^{27}$Al $^{238}$U, $^{232}$Th and $^{43}$Ca in sample SOY-19-02. Signals are in counts per seconds second (cps). This slice presents the Endo part at the bottom and Exo part at the top. The bottom part of the image corresponds to the Jurassic Kimmeridgian karst basal part. It can be easily identified by the highest signal area for $^{24}$Mg, $^{27}$Al $^{238}$U, $^{232}$Th as well as a different texture on the $^{43}$Ca mapping. The white line delimits the carbonate part of the sample.**

**(a) $^{24}$Mg**

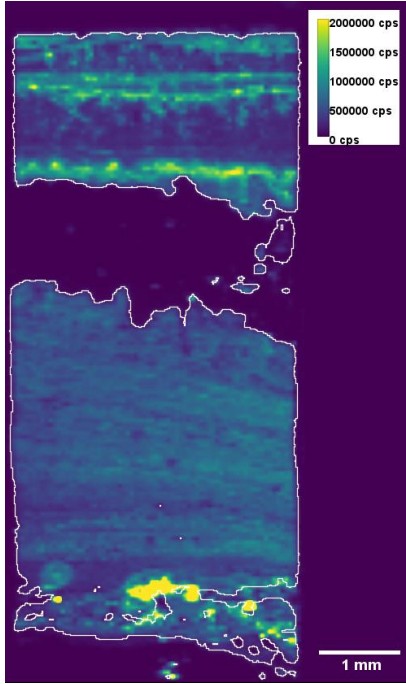

**(b) <sup>27</sup>Al**

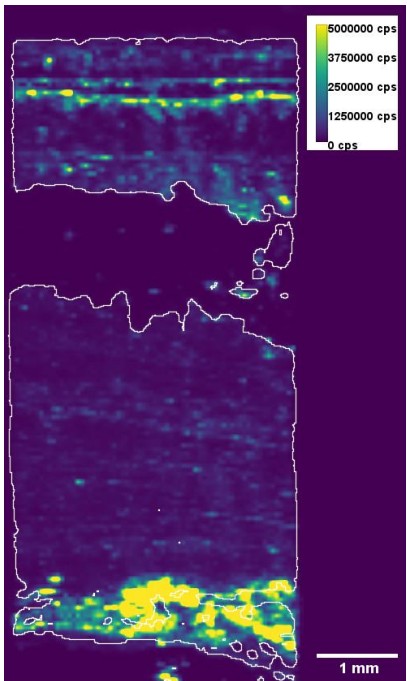


**(c) <sup>238</sup>U**

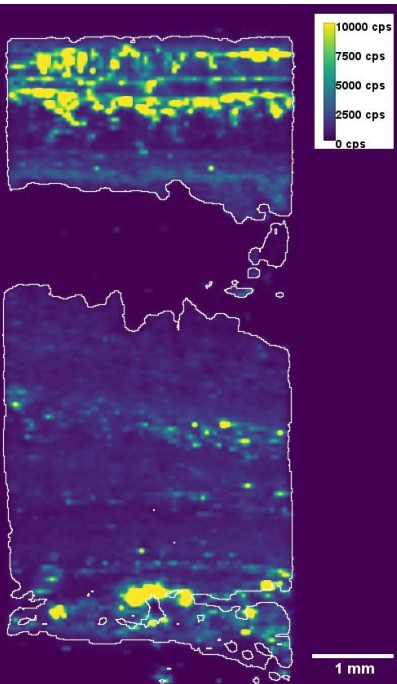

**(d)** $^{232}$**Th**

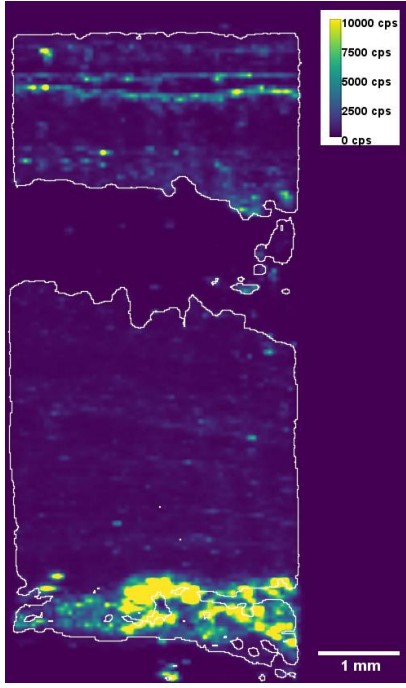

**(e)** $^{43}$**Ca**

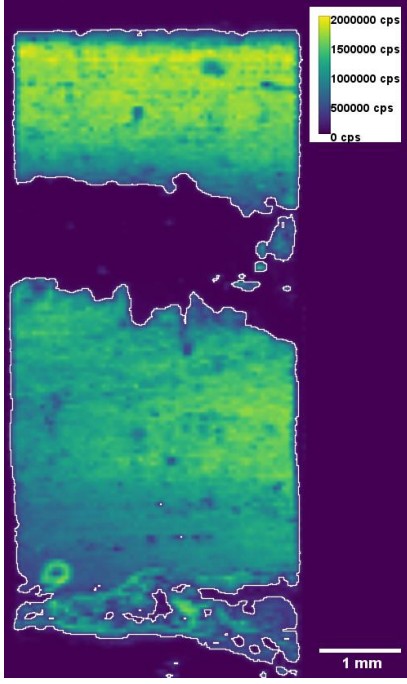

In conclusion, data from different analytical methods converge towards a coherent description of the samples. They all are composed of successive layers of pure calcite, without any evidence of diagenesis (Fig 2 and Fig 3). The detrital fraction, highlighted by the presence of Mg and Al in the fLA-single collector-ICP-SFMS images, is significant and varies between the different layers (Fig 4).

### 3.2 Dating results

The different U-Th ages obtained by Liq-MC-ICPMS and by fsLA-single collector-ICP-SFMS imaging are presented in Table 1. [14]C dating results of SOY-19-01 are given in Table 2. Figure 6 depicts all the chronological data obtained, before and after the detrital correction of the U-Th ages. All results and ages are provided with uncertainties at 95% CL.

#### 3.2.1 Liq-MC-ICPMS

[238]U contents range from $0.198 \pm 0.002$ to $0.953 \pm 0.008$ ppm with a median value of $0.495 \pm 0.004$ ppm. [232]Th contents range
from $0.257 \pm 0.002$ ppb to $140 \pm 1$ ppb with a median value of $39.4 \pm 0.3$ ppb. [230]Th/[232]Th activity ratios, from $5.3 \pm 0.2$ to $39.9 \pm 1.3$ with a median value of $17.95 \pm 0.05$, suggest that correction for detrital thorium is not negligible. The use of an *a priori* value for isotopic thorium composition of the detrital phase with a 50% uncertainty led to the notable increase of error bars for corrected ages. They range from $187.9 \pm 5.3$ ka to $1.4 \pm 0.1$ ka.

#### 3.2.2 fsLA-single collector-ICP-SFMS

The fsLA-single collector-ICP-SFMS provided qualitative mappings of [230]Th, [232]Th, [234]U and [238]U with relative variations. They confirmed the observation from sample characterization. SOY-19-02 Endo and SOY-19-01 appear relatively homogeneous, while distinct successive layers can be observed on SOY-19-02 Exo. Mappings of the [238]U signal and the [232]Th/[238]U ratio for the different samples are presented in Appendix B. These mappings were used to define "*Regions Of Interest*" (ROIs) corresponding to different calcite layers, according to Martin et al. (2022) and as shown in Fig 5. The small
variations in [238]U content and [232]Th/[238]U ratio for sample SOY19-01 did not allow such ROIs to be defined and therefore only one age was calculated. No significant difference in raw age was observed for SOY-19-02 Endo, meaning that the duration of the formation of the sample is below the age resolution of the method. Consequently, only one U-Th age was calculated for this sample using fsLA-single collector -ICP-SFMS mapping. Depending on the size, 238U content and age of the ROI, the number of [230]Th counts ranged from about $3.3 \times 10^3$ to $2.6 \times 10^5$ and the number of [234]U counts ranged from about $1.5 \times 10^4$ to
$1.0 \times 10^6$.

The detrital-rich layers identified by their Al content correspond to higher raw ages before detrital correction (Fig 6). In particular, layers 1, 3 and 7 of SOY-19-02 Exo present the highest [232]Th/[238]U ratios (Fig 5) and their uncorrected ages result in a chronological inversion with respect to the less detritus-rich layers, including the Endo part. However, when corrected for the detrital [230]Th fraction, their ages are coherent with the order of deposition of these layers using correction with an *a priori*
value of the detrital fraction, and obviously using correction assuming stratigraphic constraints (fig.6). For age corrections

using stratigraphic constraints (STRUTages routine by Roy-Barman and Pons-Branchu 2016), we assumed that initial $^{230}Th/^{232}Th$ activity ratios of the detrital phase are not constant and can vary by 30%. Note that the initial $^{230}Th/^{232}Th$ activity ratio for the detrital fraction as determined by the STRUTages routine is $1.72 \pm 0.30$, very close to the a priori value used ($1.5 \pm 0.75$).


**Figure 5: definition of the different layers of SOY-02Exo on the $^{232}Th/^{238}U$ ratio mapping obtained by fsLA-single collector-ICP-SFMS. The layer numbers are indicated on the figure, starting from 1 for the layer in contact with the Endo part to 7 for the surface layer. The ages on the left are the STRUTages computed according to Roy-Barman and Pons-Branchu (2016) and provided at 95% CL.**

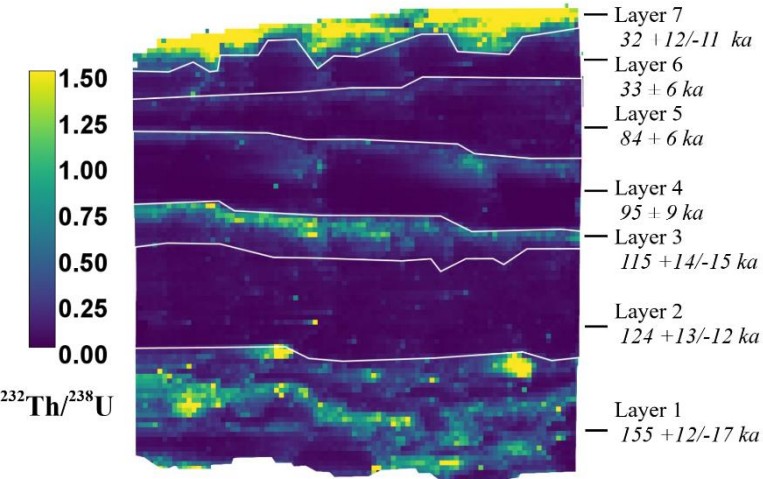


**Figure 6: Comparison of the U-Th ages obtained for samples SOY19-01 and SOY19-02 by the liq-MC-IPMS and by fsA-SC-ICP-SFMS techniques. The error bars represent the uncertainties at 95% CL.**

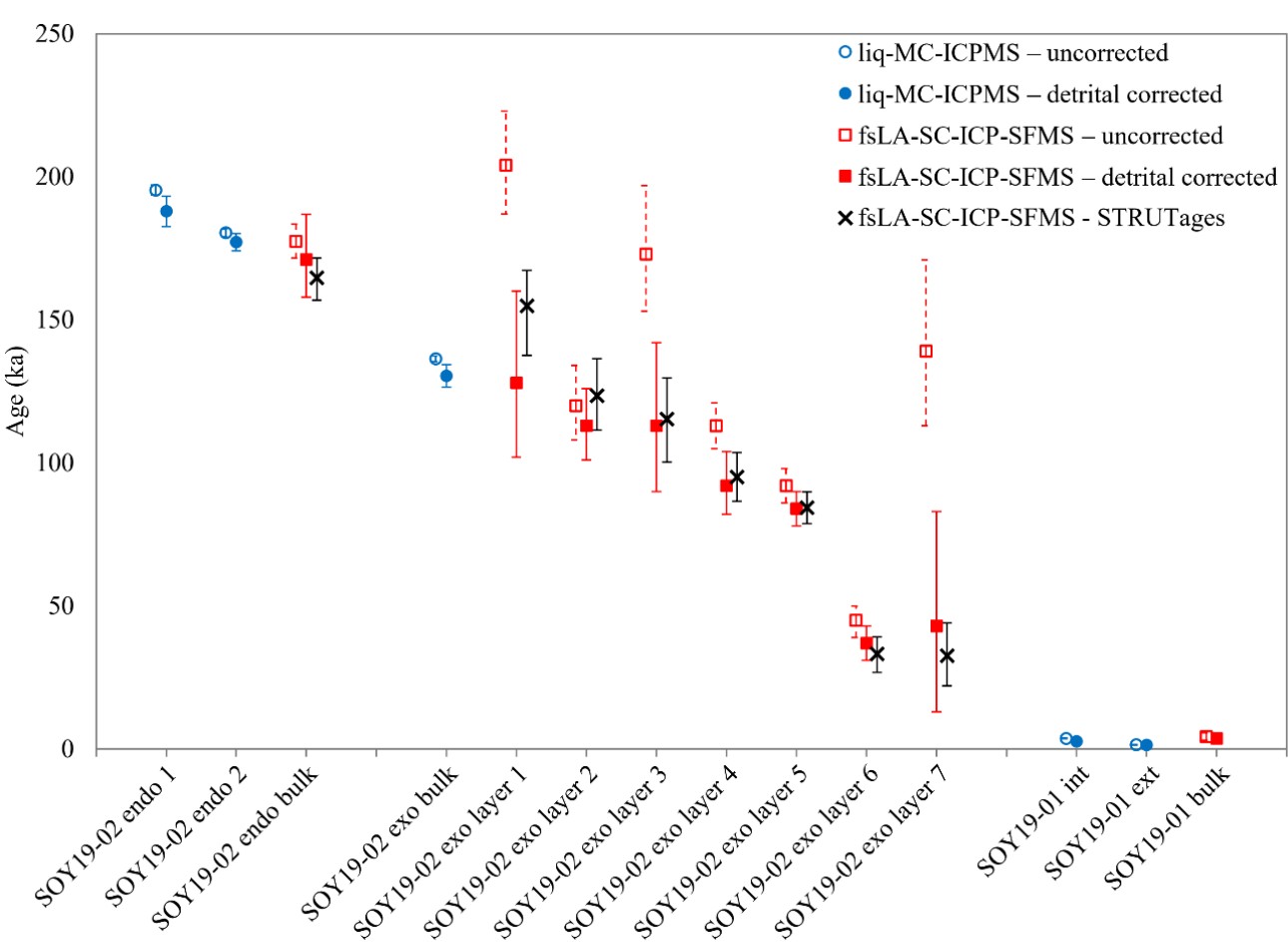


**Table 1: U-Th dating results. All uncertainties are given at 95% CL. The following half-lives were used for the calculation of ages: (4.4683±0.0024) × 10⁹ a for ²³⁸U (Jaffey et al., 1971), 245620 ± 260 a for ²³⁴U and 75584 ± 110 a for ²³⁰Th (Cheng et al., 2013).**

| Sample | Sub-sample | Method of analysis | $^{238}U$ (ppm) | $^{232}Th$ (ppb) | $\delta^{234}U_M$* (‰) | $^{230}Th/^{238}U$ ** | $^{230}Th/^{232}Th$ ** | Uncorrected Age (ka) | $\delta 234U_T$ (‰) | Detrital corrected age (ka)*** |
|---|---|---|---|---|---|---|---|---|---|---|
| SOY19-02 | Endo-1 | Liq-MC-ICPMS | 0.687 ± 0.005 | 132 ± 1 | 189.4 ± 0.9 | 1.027 ± 0.003 | 16.31 ± 0.04 | 195.3 ± 1.8 | 322.1 ± 5.1 | 187.9 ± 5.3 |
| | Endo-2 | | 0.494 ± 0.004 | 39.4 ± 0.3 | 180.8 ± 0.9 | 0.9773 ± 0.002 | 37.38 ± 0.07 | 180.4 ± 1.4 | 282.9 ± 3.1 | 177.2 ± 3.0 |
| | Endo bulk | fsLA- single collector-ICP-SFMS | 7.48 x10⁵ cps$^{†}$ | 1.37 x10⁵ cps$^{†}$ | 178 ± 8 | 0.98 ± 0.01 | 18.1 ± 0.2 | 177 ± 6 | 279 +27/-24 | 171 +17/-15 *165 +7/-8$^{††}$* |
| | Exo bulk | Liq-MC-ICPMS | 0.953 ± 0.008 | 140 ± 1 | 171.5 ± 1.2 | 0.865 ± 0.003 | 17.95 ±0.05 | 136.4 ± 1.0 | 261.4 ± 3.2 | 130.4 ± 3.9 |

| | | | | | | | | | |
|---|---|---|---|---|---|---|---|---|---|
| | Exo layer 1 | | $1.15 \times 10^6$ cps$^†$ | $4.78 \times 10^5$ cps$^†$ | $184 \pm 18$ | $1.05 \pm 0.03$ | $7.8 \pm 0.2$ | 204 +19/-17 | 269 +51/-41 | 128 +32/-26 <br> *155 +12/-17$^{††}$* |
| | Exo layer 2 | | $5.06 \times 10^5$ cps$^†$ | $2.55 \times 10^4$ cps$^†$ | $190 \pm 30$ | $0.82 \pm 0.05$ | $50.2 \pm 2.8$ | 120 +14/-12 | 262 +53/-49 | 113 +13/-12 <br> *124 +13/-12$^{††}$* |
| | Exo layer 3 | | $1.18 \times 10^6$ cps$^†$ | $4.85 \times 10^5$ cps$^†$ | $240 \pm 30$ | $1.02 \pm 0.05$ | $7.7 \pm 0.4$ | 173 +24/-20 | 334 +72/-60 | 113 +29/-23 <br> *115 +14/-15$^{††}$* |
| | Exo layer 4 | fsLA-single collector-ICP-SFMS | $1.61 \times 10^6$ cps$^†$ | $3.14 \times 10^5$ cps$^†$ | $227 \pm 20$ | $0.81 \pm 0.03$ | $12.9 \pm 0.5$ | $113 \pm 8$ | 297 +36-33 | 92 +12/-10 <br> *95 $\pm 9^{††}$* |
| | Exo layer 5 | | $2.48 \times 10^6$ cps$^†$ | $1.79 \times 10^5$ cps$^†$ | $230 \pm 18$ | $0.71 \pm 0.03$ | $30.9 \pm 1.1$ | $92 \pm 6$ | 293 +29/-28 | 84 $\pm$ 6 <br> *84 $\pm 6^{††}$* |
| | Exo layer 6 | | $8.00 \times 10^5$ cps$^†$ | $1.03 \times 10^5$ cps$^†$ | $266 \pm 36$ | $0.43 \pm 0.04$ | $10.4 \pm 0.9$ | $45 \pm 6$ | 296 +45/-44 | 37 $\pm$ 6 <br> *33 $\pm 6^{††}$* |
| | Exo layer 7 | | $3.83 \times 10^5$ cps$^†$ | $4.22 \times 10^5$ cps$^†$ | $293 \pm 58$ | $0.96 \pm 0.09$ | $2.7 \pm 0.3$ | 139 +32/-26 | 252 +111/-88 | 43 +40/-30 <br> *32 +12/-11$^{††}$* |
| SOY19-01 | int | Liq-MC-ICPMS | $0.198 \pm 0.002$ | $4.157 \pm 0.034$ | $94.9 \pm 2.0$ | $0.0366 \pm 0.0010$ | $5.3 \pm 0.2$ | $3.7 \pm 0.1$ | $95.6 \pm 2.0$ | $2.7 \pm 0.6$ |
| | ext | | $0.218 \pm 0.002$ | $0.257 \pm 0.002$ | $120.4 \pm 1.6$ | $0.0153 \pm 0.0005$ | $39.9 \pm 1.3$ | $1.5 \pm 0.1$ | $120.9 \pm 1.6$ | $1.4 \pm 0.1$ |
| | bulk | fsLA-single collector-ICP-SFMS | $2.18 \times 10^5$ cps$^†$ | $3.60 \times 10^3$ cps$^†$ | $87 \pm 23$ | $0.04 \pm 0.01$ | $8.4 \pm 2.6$ | $4.3 \pm 1.3$ | $88 \pm 23$ | $3.6 \pm 1.5$ |

\* $\delta234\,U = ([^{234}U/^{238}U]_{activity} -1).1000.$

\*\* activity ratio

\*\*\* expressed as years before 1950 (BP), detrital correction assuming a $^{230}Th/^{232}Th$ ratio of $1.5 \pm 50\%$.

$^†$in counts per second, relative quantification assuming negligible variation of the ablation rate. A quantification in mass proportion was not possible with the fsA-SC-ICP-SFMS protocol used.

$^{††}$STRUTages calculated ages (Roy-Barman and Pons-Branchu 2016) according to stratigraphic constraints (with sub-samples Endo bulk and Exo layers 1 to 7), using the STRUT ages routine). Results are given for strict coeval constraints, with 30% variability for $Ri = (230Th/232Th)_{det}$ determined here at $1.72 \pm 0.3$.

**Table 2: Results of [14]C analyses for sample Soy-19-01-int, presented as percent of Modern Carbon (pMC), uncalibrated and calibrated ages (assuming 0 and 10% of DCF) using IntCal20 atmospheric curves (Reimer et al., 2020) with OxCal 4.2 software (Bronk Ramsey, 1995). All uncertainties are given at 95% CL.**

| Lab cod | Ech | $\delta^{13}C$ | pMC | Years BP | Year Cal BP with 0% DCP | Year Cal BP with 5 % DCP | Year Cal BP with 10% DCP | Year Cal BP with 20% DCP |
|---|---|---|---|---|---|---|---|---|
| Sac - 64947 | Soy-19-01-int | -14.40 | 75.177 ± 0.207 | 2290 ± 30 | 2351 - 2181 | 1863 - 1724 | 1365 - 1301 | 541 - 510 |

### 3.3 Comparison of U-Th results obtained by the two techniques

All uncertainties in this section are given at 95% CL.

### 3.3.1 The Endo part of SOY-19-02

Two levels (internal and external part) were analysed in this thick sample using Liq-MC-ICPMS at 187.9 ± 5.3 ka and 177 ± 3.0 ka, hence a weighted mean at 179 ± 5 ka BP. Using fsLA-single collector-ICP-SFMS, a single age was calculated since the sample is very homogenous (Fig 4). The result, 163 +17/-15 or 165 +7/-8 depending on the correction used for the detrital fraction, is in agreement with the results obtained using the other technique (Table 1 and Fig 6).

### 3.3.2 The Exo part of SOY-19-02

Liq-MC-ICPMS gave a single age for the bulk sample of 130.4 ± 3.9 ka (corrected for the detrital content), whereas seven successive layers were identified and dated using fsLA-single collector-ICP-SFMS. The STRUT ages obtained for layers 1 to 5 ranged from 155 +12/-17 to 84 ± 6 ka BP (Table 1 and Fig 6). The chronological gap between layers 5 and 6 (33 ± 6 ka) may be due to drier conditions in the cave during this time interval. The small age difference between layers 6 and 7 may be due to a higher calcite deposition rate.

### 3.3.3 SOY-19-01 sample

Two levels were analysed in this thick sample using Liq-MC-ICPMS with corrected age from 2.7 ± 0.6 and 1.4 ± 0.1 ka (mean age 2.1 +1.3/-0.8ka BP). This mean age is consistent with the single value obtained using fsLA-single collector-ICP-SFMS at 3.6 ± 1.5 ka, and with the [14]C age (1.36-1.30 ky) of this sample assuming a proportion of dead carbon of up to 10% (Table 1 and Table 2).

### 3.3.4 Detrital correction

The detrital correction has a significant impact on the final U-Th ages, especially for the spatially resolved fsLA-single collector-ICP-SFMS results of the SOY-19-02 Exo part: the layers 1, 3 and 7 present considerable amounts of detrital Th associated with stratigraphic inversion of the uncorrected ages (Fig. 6, Table 1). We used two independent methods for detrital correction: a correction using an a priori value and the STRUTage model. It is noticeable that the results of these two methods

agree within the uncertainties and provide ages in stratigraphical order (Fig. 6, Table 1). This seems to indicate that the results of the two correction methods are coherent.

The accuracy of the correction by an a priori value is discussed by Hellstrom et al. (2006) because the initial detrital Th ratio $(^{230}Th/^{232}Th)A_0$ of some samples can be beyond the range covered by the a priori value of $1.50 \pm 0.75$. However, this does not seem to be the case for the sample investigated here as the correction provided coherent results and the STRUTage model calculated a compatible average value of $(^{230}Th/^{232}Th)A_0$ of $1.72 \pm 0.30$. Of course, this does not mean that this method of detrital Th correction is valid for every sample, as highlighted by Hellstrom et al. (2006), only that it is possible to check its relevance by comparing with other methods and therefore that a multi-method approach is also advisable for detrital correction in order to ensure the reliability of the dating.

The STRUTage model provides a more precise detrital corrected age than the correction by the a priori detrital value. This is made possible by the use of the stratigraphy by these models to constrain the results. Therefore, the use of this model with high spatial resolution analysis such as fsLA-single collector-ICP-SFMS imaging seems particularly appropriate and is likely to be developed further in the future. It is noticeable that the results of the STRUTage model for layer 1 of SOY-19-02 exo are significantly older than the detrital corrected ages of layer 2 and almost incompatible, within the uncertainty, while the STRUTage result for SOY-19-02 endo is significantly younger than the age determined by Liq-MC-ICPMS for the endo 2 part. This may be the result of a gap within the carbonate stratigraphy between SOY-19-02 endo and SOY-19-02 exo, corresponding to a period where the wall was covered by clay (as stated in part 2.1, a clay layer is observed between the endo and exo part of SOY-19-02 and SOY-19-03) and no carbonate was deposited at its surface. However, other explanations, such as measurement errors or a detrital Th value beyond the usual range, cannot be completely dismissed, and more work is needed to determine the cause of this potential mismatch of ages.

Martin et al. (2022) took advantage of fsLA-single collector-ICP-SFMS imaging to refine another method of detrital Th correction, the isochronal method. This method was considered for this study as part of the multi-method approach; however, it was dismissed for several reasons: the SOY-19-02 endo and SOY-19-01 detrital Th distributions were too homogeneous to provide accurate results with this method, and the areas of the same age as SOY-19-02 exo were too small to obtain enough counting statistics to be able to calculate a sufficiently precise correction. This only led to an average value of $(^{230}Th/^{232}Th)A_0$ of $1.3 \pm 0.8$, which is compatible with the a priori value used and with the average value determined by STRUTage but cannot provide precise detrital corrected ages. Although it is not appropriate for this sample, this method of detrital correction still presents a strong potential for more heterogeneous samples, or with an increase in measurement accuracy with fsLA-single collector-ICP-SFMS, by accumulating the counts over several successive imagings of the same area or sample for example, as done by Martin et al. (2022).

**3.3.5 Summary of ages obtained by comparing the two techniques**

Table 1 and Figure 6 show that the precision of the measurements made with the Liq-MC-ICPMS protocol is significantly better than those made with the fsLA-single collector-ICP-SFMS protocol. While the former is a well-established protocol

(Pons-Branchu et al., 2014), the latter is more recent (Martin et al., 2022) and further experimental work is certainly needed to improve its precision. However, most of this difference is related to the mass sampled per analysis (68 to 167 mg for the Liq-MC-ICPMS protocol compared to only 1 to 3 mg for the fsLA-SC-ICPMS protocol) and to the multicollector system used for the Liq-MC-ICPMS measurements, which offers greater counting statistics and hence greater precision in the determination of isotope ratios than a sequential system such as fsLA-single collector-ICP-SFMS. Note that the mass sample used for the Liq-MC-ICPMS protocol can also be reduced: here large pieces of samples were used because the U content was not known before the analysis. While the precision of the fsLA-single collector-ICP-SFMS ages could be increased by capturing additional images, which would therefore correspond to an increase in the mass analysed, the cost and time required to achieve the same level of precision as the Liq-MC-ICPMS protocol would be very high.

The refinement of the chronology by the STRUT ages method significantly improved the precision of the fsLA-single collector-ICP-SFMS U-Th ages, although the ages by Liq-MC-ICPMS are still more precise. Combining a high spatial resolution of ages with stratigraphical constraints has a strong potential for improving the chronological data of cave deposits. The spatial resolution of the fsLA-single collector-ICP-SFMS analysis also offers new insights into the dating of complex samples: the possibility of sampling individual layers and calculating their ages allows the rate of formation of the carbonate deposits to be estimated. On the contrary, it is noticeable that when successive layers with an age difference greater than the age resolution of the dating method are analysed in bulk, as in the case of sample SOY-19-02Exo (Table 1 and Fig 6), the apparent age obtained may not represent the average age of the layers but an average value weighted by the amounts of U-series elements present in each layer of the sub-sample. Consequently, the thickest and most U-enriched layers will have a greater weight on the apparent age compared to the other layers. Considering this, it is not surprising that the bulk age for SOY-19-02Exo obtained by Liq-MC-ICPMS seems older than what would be the average age of the individual layers of SOY-19-02Exo, as the oldest layers 1 and 2 are significantly thicker than the youngest layers 6 and 7, and therefore weigh more in the bulk age. The ability to identify the different layers for age calculation avoids this possible bias and thus improves the understanding of the calcite deposition process.

It is difficult to confirm the hypothesis that the different ROIs for U-Th dating from U and Th isotopic imaging, defined using significant variations in the $^{238}U/^{232}Th$ ratio, correspond to different periods of calcification: in some cases, there is a significant age gap between successive ROIs (for example between SOY-19-02 Exo layer 5, dated at $84 \pm 6$ ka, and SOY-19-02 Exo layer 6, dated at $33 \pm 6$ ka), and in other cases the ages of successive layers are indistinguishable within uncertainties at 95% CL. This is the case for all the initial ROIs of SOY-19-02 Endo. The simplest explanation is that the method is not precise enough to resolve the age differences between some of the ROI, which is likely considering that some of the uncertainties can be greater than 10 ka. Improving the precision via additional analysis or methodological development could enable the resolution of their ages in the future. Another possible explanation is that the real age of the ROIs with similar U-Th ages is the same, but that U and Th migrated within the calcite to form distinctive layers through diagenesis processes. However, no trace of such processes was observed in the petrography analysis nor in the U and Th isotopic mapping. Considering the very different chemical mobility of U and Th, such a migration process would likely have resulted in incoherent ages.

This last point highlights an advantage of mapping isotopes and isotope ratios for U-Th dating: they can be used to identify areas affected by detrital incorporation, and to calculate an age correction, or alternatively, to check that the usual value for detrital correction is relevant. Uranium leaching can also be highlighted, and areas affected by these changes can be excluded from the age calculation (Martin et al., 2022). As mentioned above, no trace of U leaching was observed in this study, which further supports the accuracy of the ages obtained.

In sum, it appears that the two techniques (Liq-MC-ICPMS and fsLA-single collector-ICP-SFMS) for U-Th dating are complementary, each with its own advantages and disadvantages.

## 4 Implication for rock art studies

For several years, the carbonated layers deposited on decorated walls have been dated by the U/Th method (Pike et al. 2012): depending on their location above or below the parietal representation, their dating gives a terminus ante quem or a terminus post quem for the decoration respectively. To ensure the validity of the dating results, we proposed in previous studies to use simultaneously U/Th and [14]C methods on the same carbonated sample and to compare their respective results (Plagnes et al., 2003; Valladas et al., 2017; Pons-Branchu et al., 2020). We also proposed to characterize and study the carbonate mineral and its geochemical evolution through time and to multiply the dating analysis to check the reliability of the results.

In this work, the differences between the two techniques represent a great potential to improve U-Th dating: isotope imaging and the resulting in-situ dating could indeed be used to guide the micro-sampling for the Liq-MC-ICPMS protocol. For example, for the Exo part of sample SOY-19-02, the isotope imaging indicated the need to increase the micro-sampling to better match the successive layers observed, unlike the Endo part where there were small age differences between layers.

This study illustrates a case where the samples do not show evidence of major alteration and the ages obtained with the two U-series techniques, and with [14]C, agree. The control of the basic hypothesis on which the dating is based, such as the absence of diagenesis and the application of detrital correction, reinforces the reliability and robustness of the chronology. Even in the event of alteration or disagreement on the chronology (between two different chronometers or regarding stratigraphic correction), our approach would allow the effect of alteration and the possible origins of disagreement to be investigated. This would allow errors to be corrected when possible or would indicate if any further analysis was required.

In the present study, the initial size of the samples (a few cm) is not compatible with the preservation requirements for decorated caves. However, the total mass required for petrographic analysis and dating is only a fraction of this amount, less than one gram. In the context of Le Trou du Renard cave, the initial sampling was carried out with the aim of having sufficient mass to develop a multi-method analysis and dating protocol. The development of this protocol and the data obtained now allow a more accurate assessment of the amount of sampling required to establish a reliable chronology, as well as to envisage a sampling strategy compatible with the imperative of the preservation of decorated caves. Similar analyses to this study can be carried out in non-decorated areas, or on small pieces of calcite that have fallen naturally from the decorated area and for which

partially destructive analyses can be envisaged. This would enable a chronology of calcite deposition in the cave to be

established and would highlight any difficulties for dating methods, using petrography and fsLA-single collector-ICP-SFMS. These data could then be discussed with archaeologists, geologists and conservation experts in order to identify a sampling method and the sampling points closest to the decoration that would maximize the chronological data for a minimum of sampled mass while protecting the most sensitive area of the decorations. For example, 1 to 10 mg of sample taken with a high precision micro-drill tool could then be analysed by Liq-MC-ICPMS protocol to achieve maximum precision on the age. Let's

consider a scenario where this cave presents prehistoric paintings. The analyses presented in this study can be performed on samples naturally detached from the wall, or outside the decorated area. The first and extremely important information that this study brings is that the U-Th method is reliable for dating the calcite in this site, as there is no sign of diagenesis or uranium leaching. The results from Fig.6 and Table 1 indicate that it is probably not worthwhile drilling deeper than SOY-19-02 Exo layer 5, dated as $84 \pm 6$ ka, because it is older than any cave painting discovered so far (if there are reasonable grounds for

thinking that a painting is older, then the drilling could be pursued to deeper layers). A high-precision microdrill could be mounted on an automated arm for sampling every 100µm, and the dust collected by a research-grade dust collector. This would represent about 5 mg of matter for a 0.5 mm diameter drill bit, and up to 20 mg for 1 mm diameter drill bit. These micro-samples could be analysed on site by a µXRF to check that their contents in minor elements (Mg, Al) match those expected for the layer, according to elemental mapping (Fig.4). The drilling would stop when the pigment is reached, and the micro-

samples would be prepared for U-Th dating by the Liq-MC-ICPMS protocol to achieve maximum precision on the age. The age sequences from the micro-samples, potentially improved by STRUTage (Roy-Barman and Pons-Branchu 2016) could then be inserted into the longer age sequences obtained by fsLA-single collector-ICP-SFMS. This would provide a terminus post-quem and an ante-quem for the painting, while ensuring the reliability of the ages.

## 5 Conclusion and prospective strategy

Studies carried out on prehistoric art in order to determine the cultural evolution of Palaeolithic populations require the establishment of a reliable chronology. It is thus essential to be able to control the validity of dating results obtained on carbonate parietal deposits and to check that they were not subjected to diagenesis processes.

Dating of calcite layers from the Trou du Renard cave walls (France) by the U-Th and [14]C methods revealed a long-term development of these deposits, between $187.9 \pm 5.3$ ka and $1.4 \pm 0.1$ ka.

Both U-Th dating techniques –liquid MC ICP-MS and fsLA-single collector-ICP-SFMS- agree on the chronology but lead to different precisions and resolutions. The [14]C age of the youngest sample is in agreement with the U-Th dating results. No evidence of diagenesis was detected from the petrographic (optical and cathodoluminescence microscopy and FTIR microspectroscopy) and geochemical mapping of selected chemical elements (Al, U, Th and Mg contents). The detrital incorporation of [230]Th was corrected for all U-Th ages and the dead carbon fraction was taken into account for the [14]C age.

More generally, the development of dating methods and associated technologies make it possible to use smaller sample sizes, as well as to improve the age precision and spatial resolution of analysis. These improvements allow for a better constraint and understanding of the chronology while reducing the impact of sampling on the site. Imaging methods provide geological and chemical data on samples to support the chronology, controlling several parameters such as sample homogeneity, presence of detrital phase and evidence of diagenesis (Spooner et al., 2016; Martin et al., 2022). U-Th dating using fsLA-single collector-

ICP-SFMS imaging, while not reaching the precision of liquid MC ICP-MS U-Th dating, reveals information on the chronology of the deposits with a high spatial resolution. This could contribute to assessing the chronological relationship between dated calcite deposits and archaeological remains and to constraining their relative age when a calcite deposit over/underlies them. Combining the high precision of liquid MC ICP-MS U-Th dating with imaging methods could achieve the objective of maximizing chronological information from a reduced sample size, which is a necessary condition for the

most sensitive archaeological sites. This study aimed at demonstrating the feasibility and advantages of this association.

**Appendix A: Images of the SOY-19-01 and SOY-19-03 samples in optical microscopy (plane polarized light) and cathodoluminescence (CL) microscopy**

**Figure A1: petrographic description of sample SOY-19-01. Left is optical microscopy and right is CL microscopy. Magnification is x25.**

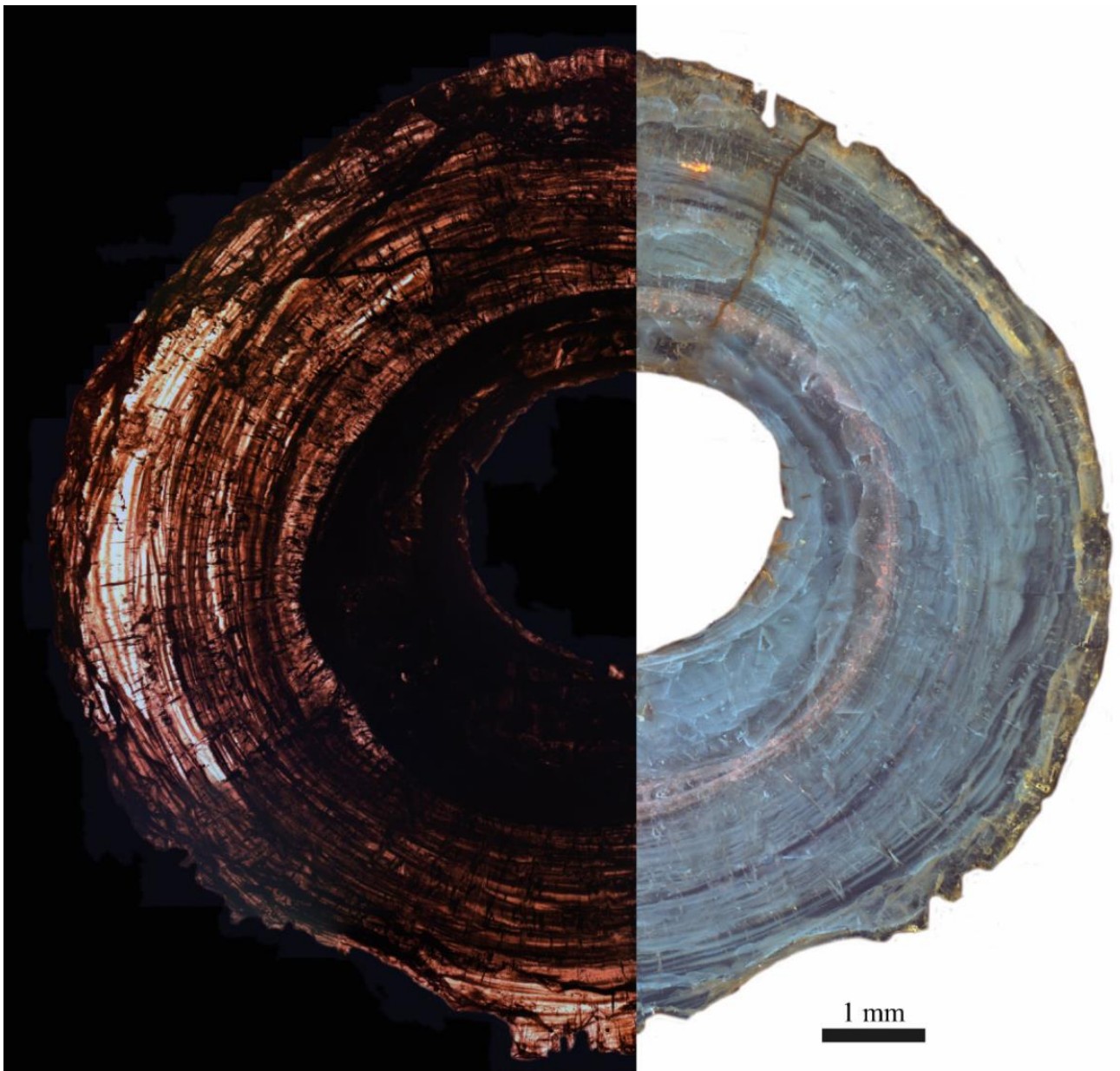

**Figure A2: petrographic description of sample SOY-19-03. Top is optical microscopy and bottom is CL microscopy Magnification is x25.**

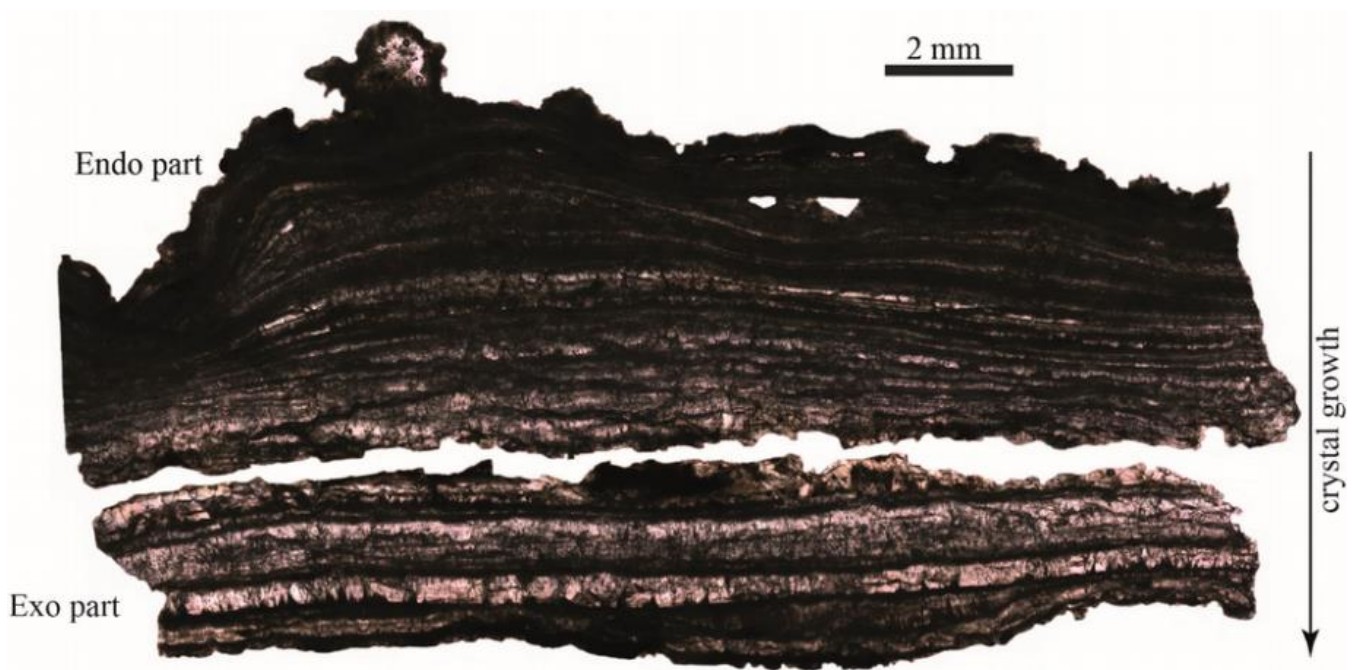

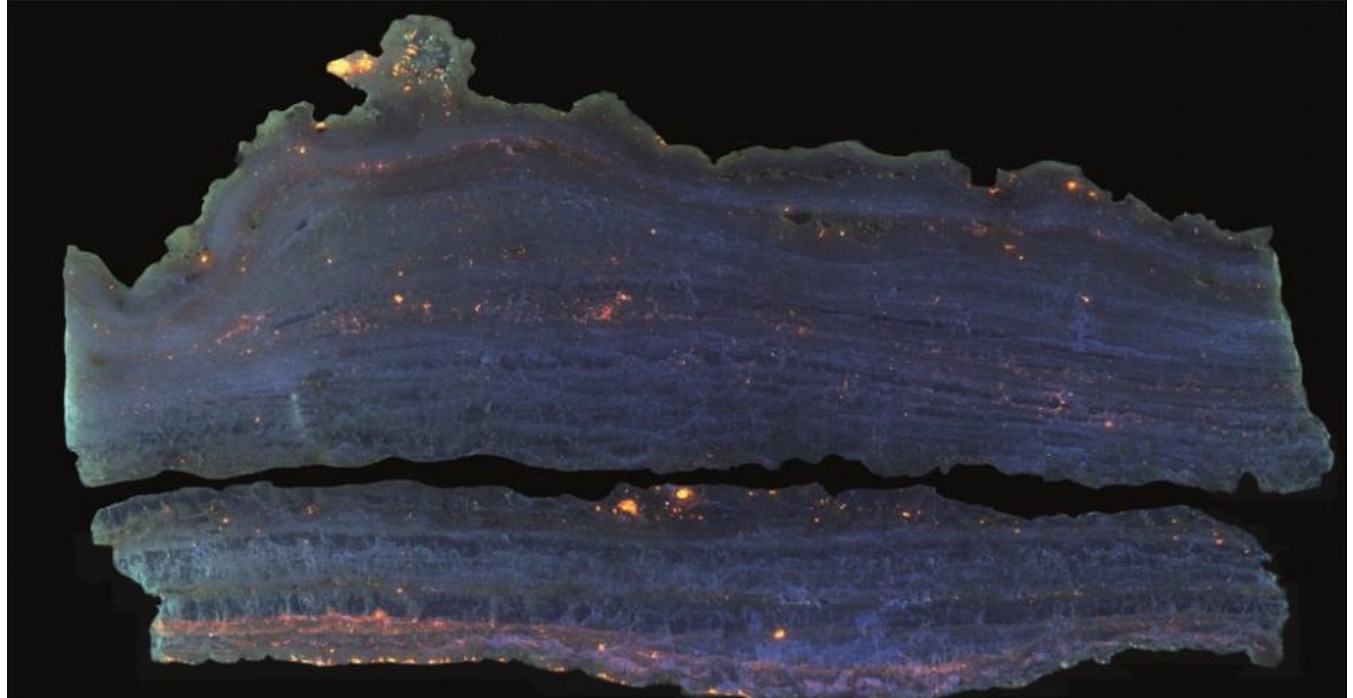

**Appendix B: fsLA-single collector-ICP-SFMS mappings of $^{238}$U signal and of the $^{232}$Th/$^{238}$U ratio for the different samples**

**Figure B1: sample SOY-19-01. The centre of the fistula is on the right of the images. The approximate positions of the sub-sampling for the Liq-MC-ICPMS protocol are indicated on (b)**

**(a) $^{238}$U**

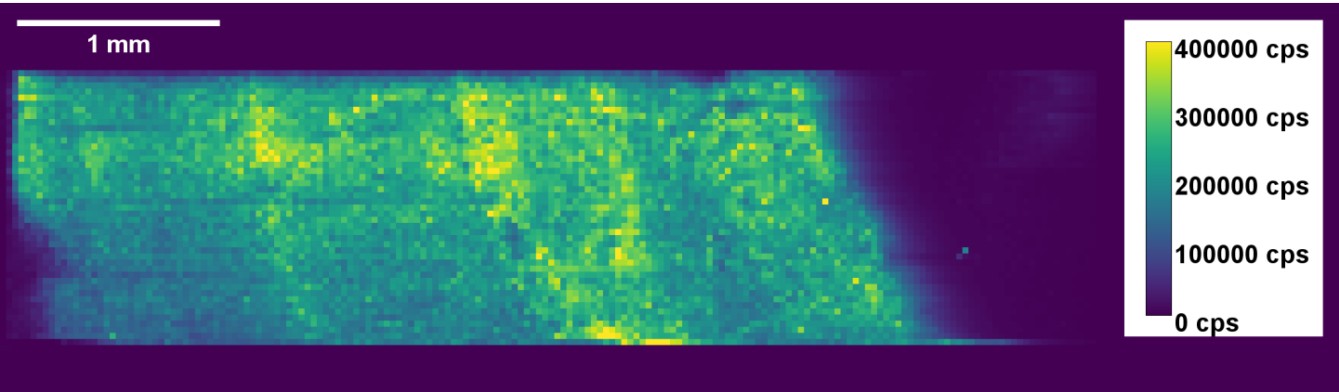

**(b) $^{232}$Th/$^{238}$U**

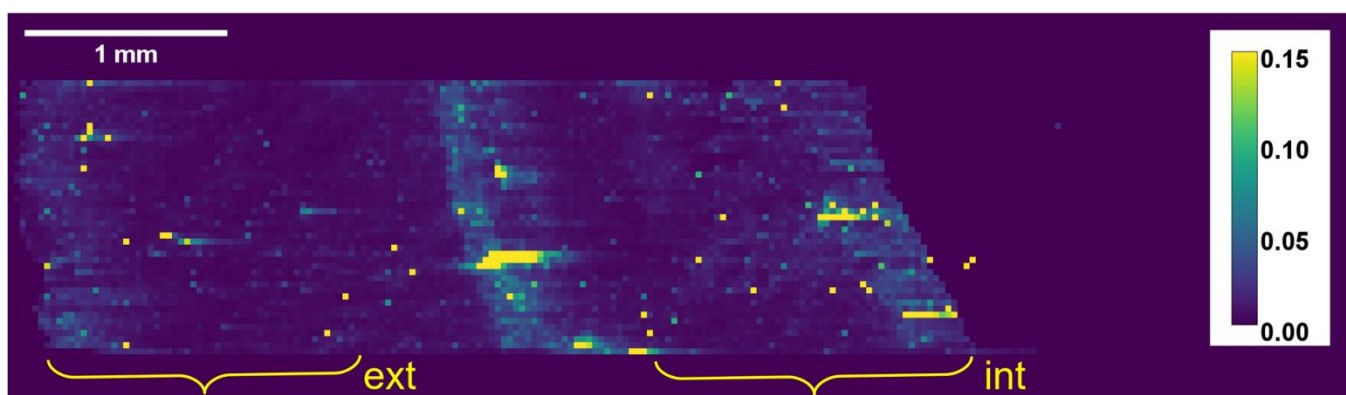

**Figure B2: sample SOY-19-02 Endo part. The growth of the calcite is from the bottom of the image to the top. The bottom part of the image corresponds to the Jurassic Kimmeridgian karst basal part. It can be easily identified on (b) by the highest $^{232}Th/^{238}U$ area. The approximate positions of the sub-sampling for the Liq-MC-ICPMS protocol are indicated on (b)**

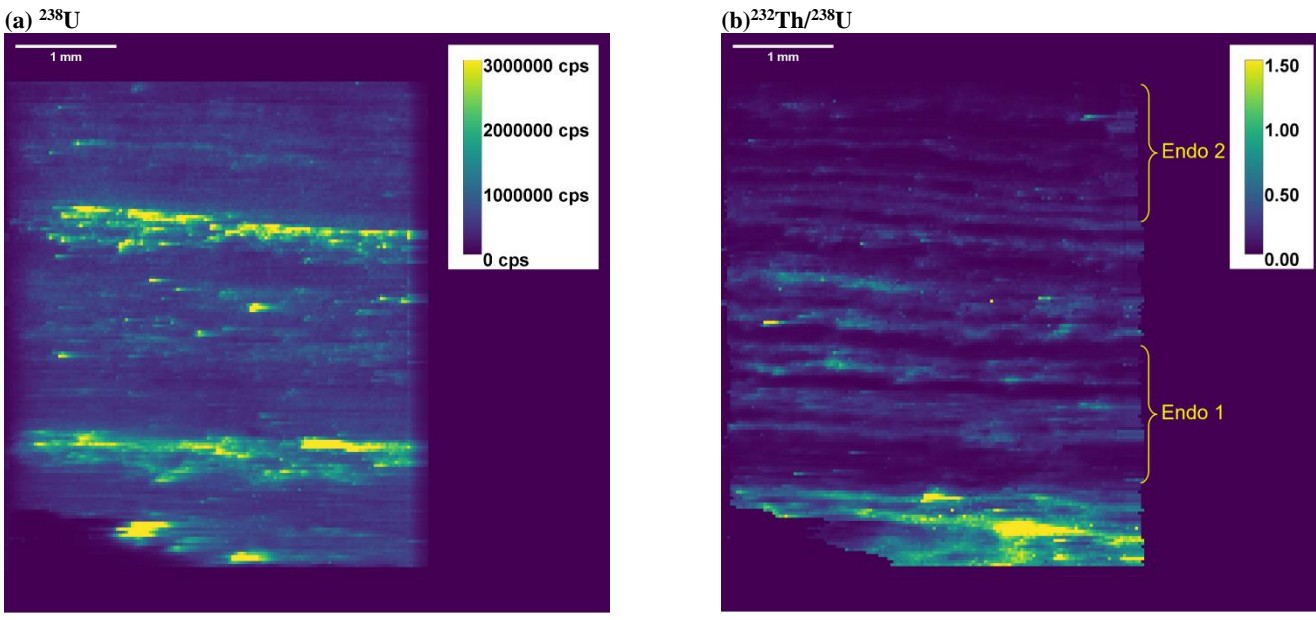

**Figure B3: sample SOY-19-02 Exo part. The growth of the calcite is from the bottom of the image to the top**

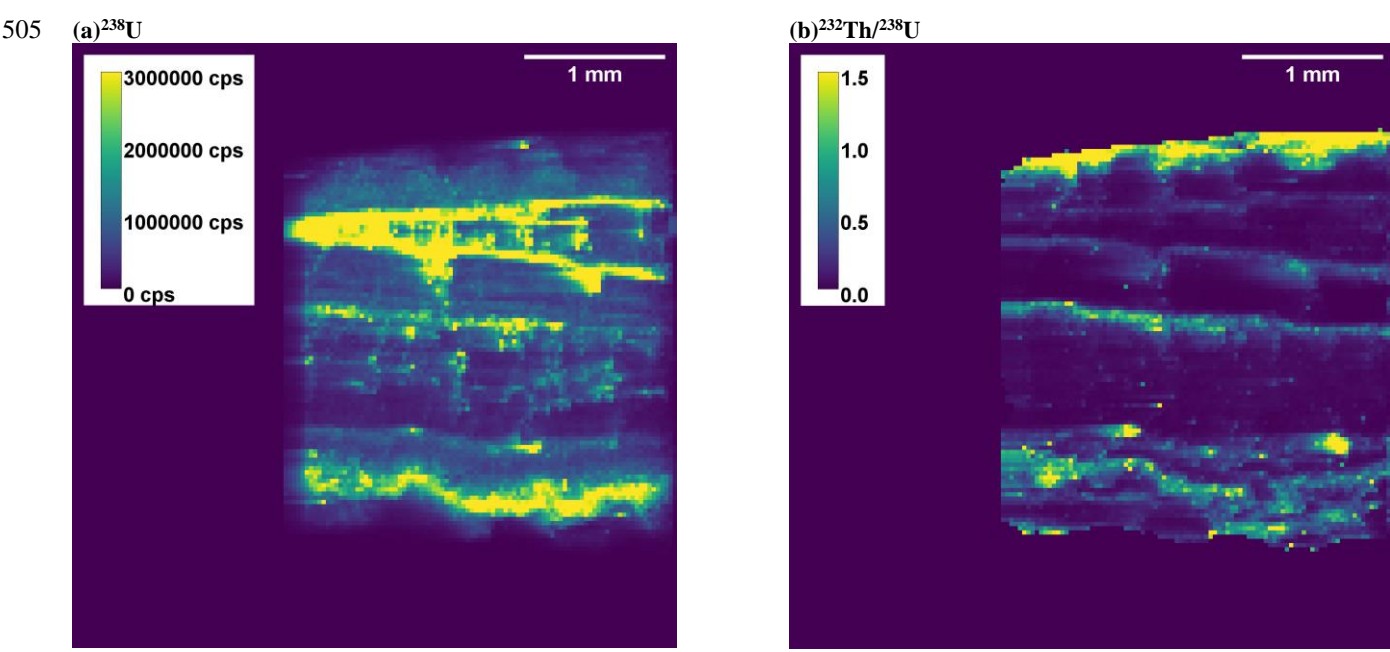

## Code availability

The STRUTages codes are available in Roy-Barman and Pons-Branchu (2016). The codes used for calculating U-Th ages from fsLA- single collector-ICP-SFMS imaging are provided in Martin et al. (2022). All codes will be made available upon request to the authors.

## Data availability

All data necessary to the study have been included. Any other data will be made available upon request to the authors.

## Sample availability

All remaining samples will be made available upon request to the authors.

## Author contribution

HV conceptualized the project, acquired funding for it and supervised it. FD and BG provided the study material. JN and BJ conducted the petrographic analysis. AD and EPB conducted the Liq-MC-ICPMS analysis. LM, GB, FC and CP performed the fsLA-ICPMS analysis under the supervision of CP. LM synthesized the data and prepared the original draft of the manuscript, which was reviewed and edited by JN, EPB, BJ, CP, NM and HV.

## Competing interests

The authors declare that they have no conflict of interest.

## Acknowledgments

This study was funded by the ANR through the APART project Grant number ANR-18-CE27-0004-01. We thank Rachel Roche Héritier of the musée archéologique of Soyons for welcoming and guiding us through the cave. The authors thank the PANOPLY analytical platform.

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
