# Peer review of "A new multi-method approach for dating cave calcite: application to the cave of the Trou du Renard (Soyons, France)"

_EGUsphere, 2023_

## Author Comment (AC2)

SOY-19-02-exo, $^{234}$U mapping (in counts per second)

[Figure]

SOY-19-02-exo, $^{230}$Th mapping (in counts per second)

---

## Author Response (AR1)

**A new multi-method approach for dating cave calcite: application to the cave of the Trou du Renard (Soyons, France)**

Loïc Martin[1,2], Julius Nouet[3], Arnaud Dapoigny[1], Gaëlle Barbotin[2], Fanny Claverie [2], Edwige Pons-Branchu[1], Jocelyn Barbarand[3], Christophe Pécheyran[2], Norbert Mercier[4], Fanny Derym[5], Bernard Gély[6], Hélène Valladas[1]

[1]LSCE/IPSL, UMR 8212, CEA-CNRS-UVSQ, Université Paris-Saclay, Chemin de Saint Aubin - RD 128, F-91191 Gif sur Yvette Cedex, France
[2]Université de Pau et des Pays de l'Adour, E2S UPPA, CNRS, IPREM, Avenue de l'Université, BP 576 64012 PAU cedex, France
[3]Géosciences Paris Saclay, Université Paris-Saclay, CNRS, bâtiment 504, 91405 Orsay, France
[4]Archéosciences Bordeaux, UMR 6034 CNRS - Université Bordeaux Montaigne
Maison de l'archéologie, Esplanade des Antilles, 33607 PESSAC Cedex
[5]Site archéologique de Soyons, 28 rue de l'église, 07130 Soyons
[6]Service Régional de l'Archéologie de la région Rhône-Alpes, 6 quai Saint Vincent 69283 Lyon cedex

*Correspondence to*: Loïc Martin (loic.martin@glasgow.ac.uk)

**Author's response to editor's review**

The authors are thankful to the editor for his comments. We tried to follow all the recommendations as much as possible for the revised version. Please find thereafter the detailed response to the comments (in blue text):

**Public justification (visible to the public if the article is accepted and published)**: I agree with both reviewers that this is an interesting and innovative paper and, with minor revision, would be publishable in Geochronology. The revisions described in the authors' responses to reviewers, including a detailed description of the analytical methodology and the STRUT algorithm used to interpret the results, as well as a detailed discussion of the detrital Th correction and its impact on the dating in section 3.3, will be welcome additions to the manuscript. Other revisions proposed by the authors will improve the manuscript's clarity. In revising this manuscript, I'd like to draw the authors' attention to several details from Reviewer 2's constructive comments and the authors' responses. These are outlined below, keyed to the line number or enumerated comment from the review/response.

- Line 129. In the revised text provided, please accompany uncertainties with whether they are reported as ±1σ or ±2σ, or otherwise. This should especially accompany important sources of uncertainty, like the measured U/Th ratio of the NIST glass.

We prefer the use of confidence level other the approximation ±1σ or ±2σ, which is more appropriate in the case of asymmetrical uncertainties like it is often the case in U/Th dating. As mentioned in our response to reviewers, all results are provided at the 95% confidence level and this has been added at the beginning of the results part. In order to avoid any confusion, we added the confidence level in the description of the results figures.

Also, please explicitly state the assumption that the U/Th elemental fractionation is the same for the glass as the carbonate analyte.

Added: "considering that laser ablation is less efficient on glass than on calcite, it is assumed that the U/Th elemental fractionation is also negligible during calcite ablation."

Additionally, the Th/U ratio reproducibility (?) of the NIST glass is given in the proposed added text as 1.00 ± 0.05 (±1σ or ±2σ is not indicated here). Should this reproducibility not be added to the uncertainty in the measured 230Th/238U ratios reported in this paper. For instance, several fsLA 230Th/238U ratios have relative uncertainties of 3%. Are these ratios more reproducible, then, than the NIST glass standard? Please provide an explanation of the 230Th/238U uncertainty budget in the
revised manuscript and/or the letter that accompanies it.

No, the U/Th ratio in the NIST is 1, and during the tuning process of the ICPMS we simply check that we measure this ratio within a 5% uncertainty, as a simple verification that the machine is running correctly. This does not mean that a 5% error is done on the U/Th ratio during measurements, and quantitative tests showing negligible U/Th fractionation have been done and the results are provided
in Martin et al. (2022). We rewritten the sentence for clarification:

"The fsLA-ICPMS coupling was tuned daily with a NIST 612 glass sample in order to obtain the best sensitivity while ensuring a complete atomisation of the particles. This was achieved by checking that the value of the U/Th ratio measured on the NIST 612 corresponded to the reference value of 1 ± 0.05 at 95% confidence level (95% CL)."

- Line 132. Please evaluate, in the Discussion section, whether the different ROIs correspond to different periods of calcification as requested by the reviewer.

A paragraph has been added in section 3.3.5 :

"It is noticeable that the hypothesis that the different ROIs for U-Th dating from U and Th isotopic
imaging, defined using significant variation of the 238U/232Th ratio, correspond to different period of calcification is difficult to confirm: in some cases, there is a significant age gap between successive ROIs (for example between SOY-19-02 Exo layer 5, dated at 84 ± 6 ka, and SOY-19-02 Exo layer 6, dated at 33 ± 6 ka), and in other case the ages of successive layers are indistinguishable within uncertainties at 95% CL, like it is the case for all initial ROIs of  SOY-19-02 Endo. The simplest explanation is that the precision
of the method is not enough to resolve the ages difference between some of the ROI, which is likely considering that some of the uncertainties can be other 10 ka. Improvement of the precision via additional analysis or methodological development could enable the resolution of their ages in the future. Another possible explanation is that the real age of the ROIs with similar U-Th ages is the same, but U and Th migrated within the calcite to form distinctive layers through diagenesis processes.
However, no trace of such process has been observed in the petrography analysis nor in the U and Th isotopic mapping. Considering the very different chemical mobility of U and Th, such a migration process would likely have resulted in incoherent ages."

- Line 178. In discussion of homogeneity, please provide context for qualitative descriptors like "good"
with a quantitive metric ("varies by a factor of less than two") or some additional context taken from published literature.

We clarified this passage and added some quantitative metrics:

"A large Al and Mg rich zone is noticeable at the root of SOY19-02 Endo (at the top of the images) on the fsLA-single collector ICP-SFMS mappings, which corresponds to a piece of the limestone host rock mixed with calcite deposit. Apart from this basal part, the fsLA-single collector ICP-SFMS mappings of SOY19-02 Endo indicate a significantly more homogeneous distribution of the chemical elements investigated (24Mg, 27Al, 238U, 232Th and 43Ca) than SOY19-02 Exo, which shows standard deviation between pixel values 1.3 to 3 times higher than SOY19-02 Endo."

- Figure 1. I also had difficulty in confidently separating the endo and exo layers. This is more difficult for readers who are not familiar with your samples or with carbonate coatings. One solution might be to provide a sample sketch beside your photograph with clearly labelled extents of each. GChron does not charge for extra figures or subfigures.

    A Delimitation between the endo part and the exo part has been added to the figure.

    - Figures, enumerated point 3) I agree with reviewer 2 here that it is difficult to tell where the edge of the samples are in the figures. This comment is not about color ramps or saturation, but about image interpretation. For instance, in Figure 4, is there a low 232Th region at the top of the image, or is that just the edge of the sample? How does it correlate spatially with the higher 238U concentrations nearby? To find out, I need to squint at the 43Ca image and try and imagine the upper boundary of that image on the 232Th map. A solid white line around the top, bottom, and sides of the sample edges, as described e.g. by the 43Ca maps, would be quite helpful in this regard, or at least worth exploring.

    We added a solid line on Fig.4 for delimitating the carbonate part of the sample.

- Figures, enumerated point 5) I look forward to seeing the revised version of the figure, but would also suggest "jittering" the data with small horizontal offsets to improve readability. I (and doubtless the reviewer) realize that the x-axis is not the real number line, but that doesn't mean that you can't budge the points' uncertainty bars horizontally so that they don't all plot on top of one another.

    Done. With the jittering, we went back to the previous color code for simplification and clarity.

    - Line 211. Please add median values as requested by the reviewer.

    The median values have been added.

    - Line 214. Is this (95% confidence intervals for all qualitative results) true for measured and assumed uncertainties as well? Please be clear throughout the manuscript, and when in doubt, indicate the confidence level directly. That way, a reader need not go searching through your text for the correct confidence level.

    Yes, all data are presented with uncertainty at 95% confidence level. This has been indicated at different point through the manuscript as well as in the figure caption and table caption when presenting quantitative results. See also answer to comment about line 129.

    - Lines 220-224. Please carefully address the point by reviewer 2 about preparation of the "bulk" sample for SOY19-02. This remark is important but is dismissed as "self-evident" in the authors' response.

We added all the process of preparation of the SOY19-02 "bulk" in section 2.4.1, but beyond the cleaning of surface clay and the fact that the bulk contain all the layers of this sub-sampled (which is the usual meaning of the term bulk), there is nothing more.

    - Table 1, comments about three vs. seven ROIs of interest. In their response, the authors do not
address whether the exo part of SOY-19-02 deserves to get three or seven ROIs given that several of the seven layers do not have resolvable ages. This important interpretative choice is of interest to others looking to apply this technique (i.e., an important audience of this paper), and could use a few sentences in the revised manuscript.

    A discussion paragraph has been added in section 3.3.5, see answer to the comment about line 132.

    - Line 293. Given that data rejection based on hypothesized uranium leaching is not part of this study, it can safely be excluded from this manuscript. It is already discussed in detail in Martin et al. 2022 and need not be duplicated here.

    A part of this manuscript concerns the checking methods to ensure that the obtained U-Th ages are
representative of the age of the sample. For example, the petrology analyses are checking that no sign of diagenesis, which could have questioned the ages, are present. In the same perspective, checking the absence of U leaching is a necessary step to ensure that the U-Th age are reliable, which is very important for the application of the method to major archaeological sites and therefore needs to be reminded. It is also used in the newly added paragraph discussing of the relevance of some ROIs in
regards of their similar ages, to infirm the possibility of U migration within a same age layer. Because of these two points, we insist that the mention of uranium leaching is necessary for this study.

    - Line 324. Please make sure that the issues raised here are clearly addressed in the revised version of this manuscript. Specifically, outline the scenario I think you allude to (I'm not totally sure from the text
or the response) where larger samples are taken farther from the decorated areas, characterized by laser ablation, and then smaller samples are recovered closer to the decorated area for… solution MC-ICPMS                                  U-Th                                    analysis?
    Further details and examples have been added to part 4 to answer the question raised by reviewer 2. A whole paragraph detailing the potential scenario for sampling in q decorated cave has been added
to part 4.

    Additionally, please address my own small list of suggestions and minor edits:
    - Line 97. Here and elsewhere, please make sure the -1 is superscripted in cm-1.

    Done.

    - Line 109. Spell out numbers less than or equal to ten in text, like "…by two galvanometric…"

    Done.

    - Line 111. Here and throughout the manuscript, add a space between values and their units, like 50
µm, 50 µm.s-1, and 1 s.

    Done.

- Table 1: Provide column info in footnotes instead of referencing numbered columns.

Done

- Table 1: Indicate in the table whether the given uncertainties are ±1σ or ±2σ, preferably in all appropriate column headings.

All data are presented with uncertainty at 95% confidence level, this has been added in the Table caption.

- Table 1: Spell out "interior" and "exterior" for SOY19-01 sub-samples.

SOY19-01 int and SOY19-01 ext have been defined in part 2.4.1 and on Fig.B1 in the revised manuscript, therefore it is more coherent to let "int" and "ext" in Table1.

- Table 1: Please provide an estimate of 238U ppm and 232Th ppm for fsLA. I see your discussion of ablation rates, etc, but it's ok if this estimate is not made to the same precision as the isotope ratios.

Please understand that this is not a question of precision: the quantification of $^{238}$U and $^{232}$Th cannot be made because the data necessary to do it (the counts rate on the $^{43}$Ca to estimate the ablation rate and a specific calibration on a solid carbonate reference sample for quantification of these two element) were not acquired during the analysis. The reason is that they are not necessary for the accurate determination of isotopic ratio with laser ablation ICPMS, measuring the $^{43}$Ca would reduce the accuracy of measurements of the other element (because the time spent on measuring this element is not spent on the other elements) and we used a system of liquid calibration system which does not calibrate for the ablation rate.

With the liquid protocol, it is always possible to calculate the $^{238}$U and $^{232}$Th content because the use and measurement of spike allows an internal calibration for U and Th, but it is not the case with laser ablation ICPMS. We added the count rate measured for $^{238}$U and $^{232}$Th, which can be a proxy to estimate the variation between the different layers (assuming that the ablation rate does not vary significantly) but this cannot be considered in any case as a proper quantification of the contents.

- Table 1: Please don't use color in highlighting rows.

The color highlight has been replaced by grey.

- Line 258. Change "7" to "seven"

Done.

- Line 273. Indicate which (make/model) "multicollector system" was used.

This information was given in the methodology part, section 2.4.1. We added the make: "Thermo Scientific™ Neptune ™ Plus"

- Figure B2: There are two Figure B2s.

Corrected.

- Second Figure B2: Change "if" to "of".

Done.

---

## Author Response (AR2)

Technical Corrections:

In the Table 1 caption, after reporting that all uncertainties are 95% CL, the Jaffey et al. 238U decay constant is given at ±1sigma. Also, when using the Jaffey et al. and Cheng et al. decay constants together, especially to construct age models, you can use the smaller set of uncertainties reported in Cheng et al. to avoid double-propagating the (systematic) 238U decay constant uncertainty into the 234U and 230Th decay constant uncertainties. I recognize the decay constant uncertainties are not important for this application, though, and you may remove all of them from the manuscript if you prefer.

Response: We are thankful to the editor for noticing this mistake, we corrected into a 2 sigma uncertainty for the 238U decay constant. All the calculation were done without using the reduced set of uncertainties of Cheng et al. (2013). While we recognize that using this set would be more correct, it has no significant change on the age uncertainties reported in this study, so at that stage of the publication we prefer to keep it as it is. We also prefer to keep in the manuscript the uncertainties of the decay constant that we used in the calculation for transparency purpose.

In the Table 1 footnote ***, change "ratio" to "activity ratio" or enclose the ratio in parentheses.

Response: Done

Figure 5 caption: The ages on the *right* are the STRUTages…

Response: Done

Line 148: Add a period after chamber.

Response: Done

Line 154: Remove the acronym "SRM" that does not apply to IRMM-184 and add a reference to the 2022 Richter et al. recertification of IRMM-184. Also, note that its estimated 235U/238U atom ratio changed to 0.0072631 ± 0.0000011 (95% CI). The change will make no difference to your reported data and ages, but you might as well cite the most up-to-date value. You can find that report here:
https://crm.jrc.ec.europa.eu/p/IRMM-184/n-236U-n-238U-uranium-isotope-amount-ratio-5-M-HNO3-solution-HNO3-n-235U-n-238U-n-234U-n-238U/IRMM-184-URANIUM-238-NATURAL-ISOTOPIC-NITRATE-SOLUTION/IRMM-184

Response: Done

Line 161: Change 'value' to 'activity ratio'.

Response: Done

Line 162: Change 150 to 1.50. The *** subscript in Table 1 gives detrital 230Th/232Th activity ratio uncertainty as ± 0.50 and here in the text is ± 0.75. Please change the relevant uncertainty in the text or footnote and double-check that it matches the uncertainty propagated into the appropriate Table 1 detrital-corrected ages.

Response: Done. We replaced the 1.50 ± 0.50% uncertainties in Table 1 subscript by 1.50 ± 0.75 .

Line 389: Change 'diagenesis' to 'diagenetic'.

Response: Done

Line 390: Change 'such processes' to 'diagenesis' and 'petrography' to 'petrographic'.

Response: Done

Line 429: Change "Let's consider a scenario where this cave presents historic paintings. The analysis presented in this study can be performed…" to "Considering a cave with historic paintings, the analyses presented in this study could be performed.."

Response: Done

Please notice that there was a mistake in the reported masses in the sentence line 435:

"A high-precision microdrill could be mounted on an automated arm for sampling every 100 μm, and the dust collected by a research-grade dust collector. This would represent about 5 mg of matter for a 0.5 mm diameter drill bit, and up to 20 mg for 1 mm diameter drill bit"

This sentence has been replaced by the following, with corrected values:

"A high-precision microdrill could be mounted on an automated arm for sampling every 200 μm, and the dust collected by a research-grade dust collector. This would represent about 3 mg of matter for a 2.5 mm diameter drilling, and up to 11 mg for 5 mm diameter drilling".